# Sigma oscillations protect or reinstate motor memory depending on their temporal coordination with slow waves

Judith Nicolas[1,2]*, Bradley R King[3], David Levesque[4], Latifa Lazzouni[5], Emily Coffey[6], Stephan Swinnen[1,2], Julien Doyon[5], Julie Carrier[4,7], Genevieve Albouy[1,2,3]*

[1]Department of Movement Sciences, Movement Control and Neuroplasticity Research Group, Leuven, Belgium; [2]LBI - KU Leuven Brain Institute, KU Leuven, Leuven, Belgium; [3]Department of Health and Kinesiology, College of Health, University of Utah, Salt Lake City, United States; [4]Center for Advanced Research in Sleep Medicine, Centre Intégré Universitaire de Santé et de Services Sociaux du Nord-de-l'Ile de Montréal, Montreal, Canada; [5]McConnell Brain Imaging Centre, Department of Neurology and Neurosurgery, Montreal Neurological Institute, McGill University, Montreal, Canada; [6]Department of Psychology, Concordia University, Montréal, Canada; [7]Department of Psychology, Université de Montréal, Montreal, Canada

*For correspondence:
nicolasjdh@gmail.com (JN);
genevieve.albouy@kuleuven.
be (GA)

Competing interest: The authors declare that no competing interests exist.

**Abstract** Targeted memory reactivation (TMR) during post-learning sleep is known to enhance motor memory consolidation but the underlying neurophysiological processes remain unclear. Here, we confirm the beneficial effect of auditory TMR on motor performance. At the neural level, TMR enhanced slow wave (SW) characteristics. Additionally, greater TMR-related phase-amplitude coupling between slow (0.5–2 Hz) and sigma (12–16 Hz) oscillations after the SW peak was related to higher TMR effect on performance. Importantly, sounds that were not associated to learning strengthened SW-sigma coupling at the SW trough. Moreover, the increase in sigma power nested in the trough of the potential evoked by the unassociated sounds was related to the TMR benefit. Altogether, our data suggest that, depending on their precise temporal coordination during post learning sleep, slow and sigma oscillations play a crucial role in either memory reinstatement or protection against irrelevant information; two processes that critically contribute to motor memory consolidation.

## Editor's evaluation

The authors demonstrate that targeted memory reactivation (TMR) can enhance motor memory consolidation. TMR has mainly been used to strengthen declarative memories, and, on a neurophysiological level, TMR has been shown to strengthen oscillatory cross-frequency coupling. Here the authors extend previous findings into the motor domain and reveal that TMR strengthens motor memories and again, cross-frequency coupling of cardinal sleep oscillations, namely slow waves and spindles. Collectively, their findings provide additional evidence for the idea that the temporal precision of cross-frequency network coordination is critical for timed information transfer from short-term to long-term mnemonic storage.

## Introduction

Motor memory is the capacity that affords the development of a repertoire of motor skills essential for daily life activities such as typing on a keyboard or buttoning a shirt. After initial acquisition, a motor memory undergoes consolidation, which is the offline (i.e. without further practice) process by which

the acquired memory trace becomes stable and long-lasting (*Maquet, 2001*; *Robertson et al., 2004*). Sleep, and non-rapid eye movement sleep (NREM) in particular (*Albouy et al., 2008*; *Albouy et al., 2013*), is thought to offer a privileged window for the consolidation process to occur (*King et al., 2017*). The specific electrophysiological events characterizing NREM sleep, such as slow waves (SW - high amplitude waves in the 0.5–2 Hz frequency band) (*Ngo et al., 2013b*), thalamo-cortical spindles (short burst of oscillatory activity in the 12–16 Hz sigma band) (*Barakat et al., 2013*; *Ngo et al., 2019*) and hippocampal ripples (80–100 Hz oscillations in humans) (*Axmacher et al., 2008*), as well as their precise synchrony, have been described to support neuroplasticity processes underlying consolidation (*Muehlroth et al., 2019*).

In recent years, there has been increasing evidence in both the declarative and motor memory domains that consolidation processes can be augmented by experimental interventions such as targeted memory reactivation (TMR) applied during post-learning sleep (*Rudoy et al., 2009*; *Cousins et al., 2016*; *Schönauer et al., 2014*; *Hu et al., 2020*). In TMR protocols, sensory stimuli (e.g. sounds) that are associated to the learned material during the learning episode are presented offline, during the consolidation interval, in order to reactivate the encoded memory trace (*Ngo et al., 2013b*). This memory reinstatement is thought to be supported by a TMR-mediated reinforcement of the endogenous brain reactivation patterns that occur spontaneously during the consolidation process. Such reactivations are thought to support the transfer of memory traces to the neocortex (*Born and Wilhelm, 2012*). While the beneficial effect of TMR on motor performance has been highlighted in previous research (e.g. *Antony et al., 2012* ; *Faul et al., 2007*; *Schönauer et al., 2014*; *Cousins et al., 2016*), the neurophysiological processes supporting these effects have been scarcely studied. Therefore, the goal of the present study was to elucidate the neurophysiological processes supporting memory reactivation during sleep which underlie TMR-induced enhancement in motor memory consolidation.

We designed a within-participant experiment (*Figure 1*) pre-registered in the Open Science Framework (available at *https://osf.io/y48wq*). Young healthy participants were trained on a Serial Reaction Time task (*Nissen and Bullemer, 1987* ) during which they learned two different motor sequences, each associated to a particular sound. Participants were then offered a 90-min nap that was monitored with polysomnography. During NREM 2–3 sleep stages, the sound associated to one of the two trained sequences ('Associated' sound to the 'Reactivated' sequence) as well as a control sound ('Unassociated') that was not associated to the learned material were played. The sound associated to the other learned sequence was not presented during the nap, thus serving as a no-reactivation control condition ('Non-reactivated'). The time course of the TMR-induced consolidation process was assessed with retests after the nap episode as well as after a night of sleep spent at home. At the behavioral level, results demonstrated the expected TMR benefit. At the brain level, they indicate a TMR-mediated enhancement of SWs and SW-sigma coupling after the peak of the SW such that the higher the coupling, the greater the effect of TMR on motor performance. Intriguingly, unassociated sounds also strengthened SW-sigma coupling but at a different phase of the SW (trough) and the increase in sigma power nested in the trough of the potential evoked by the unassociated sounds was related to the TMR benefit. Altogether, our findings suggest that sigma oscillations may play a dual role in the consolidation process depending on both the nature of the information to be processed and the phase of the slow oscillation in which they occur. We propose that sigma oscillations protect or reinstate motor memory depending on their temporal coordination with slow oscillations during post-learning sleep.

## Results

The analyses presented in the current paper that were not pre-registered are referred to as *exploratory*.

### Behavioral data

As per our pre-registration, behavioral analyses focused on performance speed (i.e. response time (RT) on correct key presses) on the motor sequence learning task measured at three time points: pre-nap, post-nap, and post-night (*Figure 1a*). Results of the analyses on performance accuracy are presented in *Figure 2—figure supplement 1*.

Analyses of the pre-nap training data indicated that participants learned the two sequence conditions (reactivated and non-reactivated sequences) to a similar extent during initial learning (16 blocks

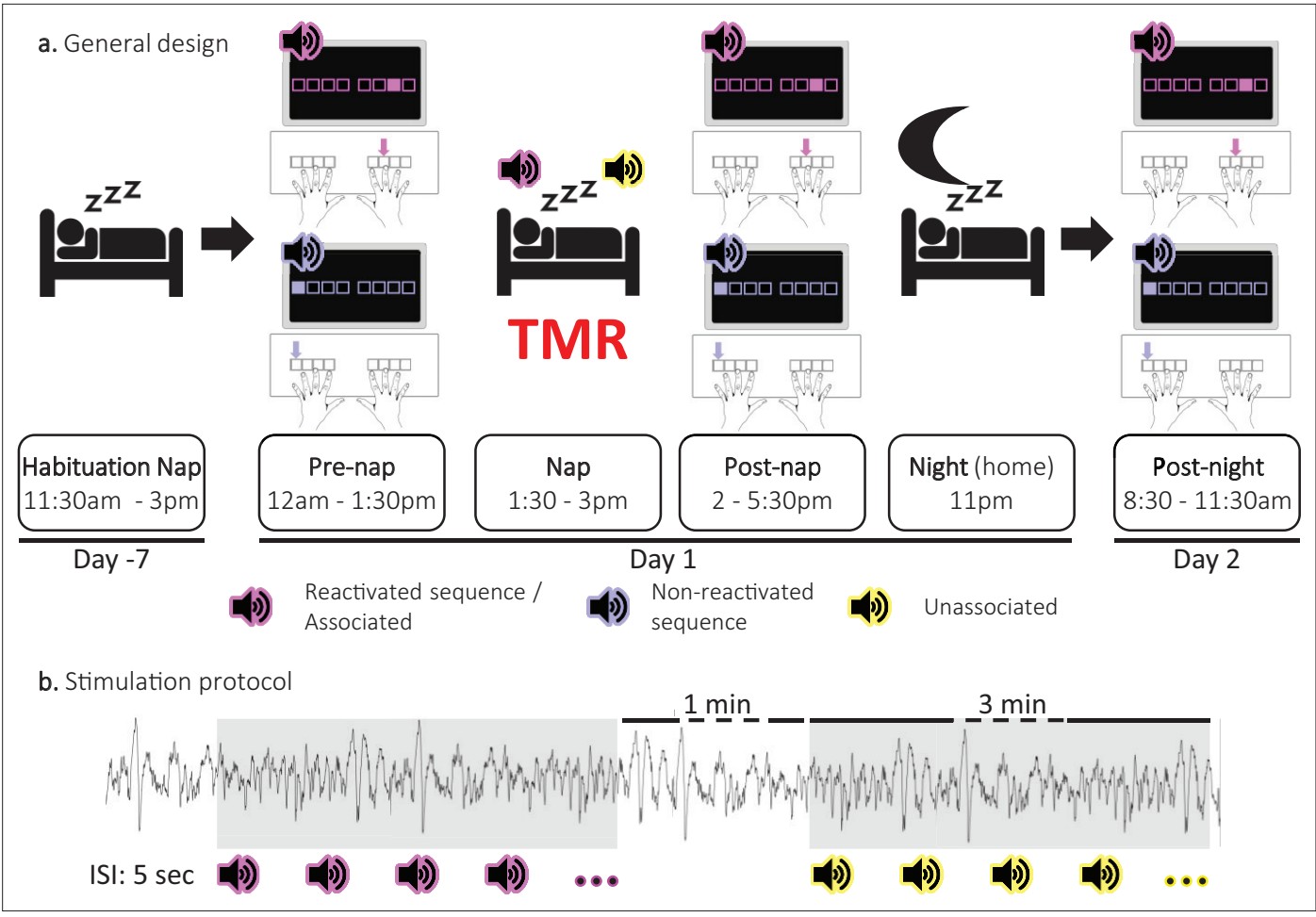

**Figure 1.** Experimental protocol. (**a**) General design. Following a habituation nap that was completed approximately one week prior to the experiment, participants underwent a pre-nap motor task session, a 90 minute nap episode monitored with polysomnography during which targeted memory reactivation (TMR) was applied and a post-nap retest session. Participants returned to the lab the following morning to complete an overnight retest (post-night). During the motor task, two movement sequences were learned simultaneously and were cued by two different auditory tones. For each movement sequence, the respective auditory tone was presented prior to each sequence execution (i.e. one tone per sequence). One of these specific sounds was replayed during the subsequent sleep episode (**Reactivated**) and the other one was not (**Non-reactivated**). During the NREM 2–3 stages of the post-learning nap, two different sounds were presented. One was the sound associated (**Associated**) to one of the previously learned sequences, that is to the reactivated sequence, and one was novel, that is not associated to any learned material (**Unassociated**). (**b**) Stimulation protocol. Stimuli were presented during three-minute stimulation intervals of each cue type alternating with a silent 1 minute period (rest intervals). The inter-stimulus interval (**ISI**) was of 5 sec. The stimulation was manually stopped when the experimenter detected REM sleep, NREM1 or wakefulness.

of training; main effect of Block: F(15, 345)=34.82; p-value = 2.04e-26; $\eta^2$=0.6; main effect of Condition: F(1, 23)=0.16; p-value = 0.69; Block by Condition interaction: F(15, 345)=1.09; p-value = 0.37; *Figure 2a*). After initial training, participants were offered a short break (~5 min) and were then tested again on the learned motor sequences. This short pre-nap test session was designed to offer a fatigue-free measure of the end-of-training, asymptotic performance to be used as baseline for the computation of subsequent offline changes in performance (see description below) (*Pan and Rickard, 2015*). Before computing offline changes in performance, we first assessed whether participants reached stable and similar performance levels between conditions during the pre-nap test session. Results showed that while performance reached similar levels between conditions (4 blocks; main effect of Condition: F(1,23) = 3.39e-5; p-value = 0.99; Block by Condition interaction: F(3,69) = 1.21; p-value = 0.31), asymptotic performance levels were not reached as shown by a significant Block effect (F(3,69) = 6.67; p-value = 0.001; $\eta^2$=0.22). To meet the performance plateau pre-requisite to compute offline changes in performance, the first block of the pre-nap test session driving this effect was removed from further analyses. Performance on remaining blocks was stable as indicated by a non-significant

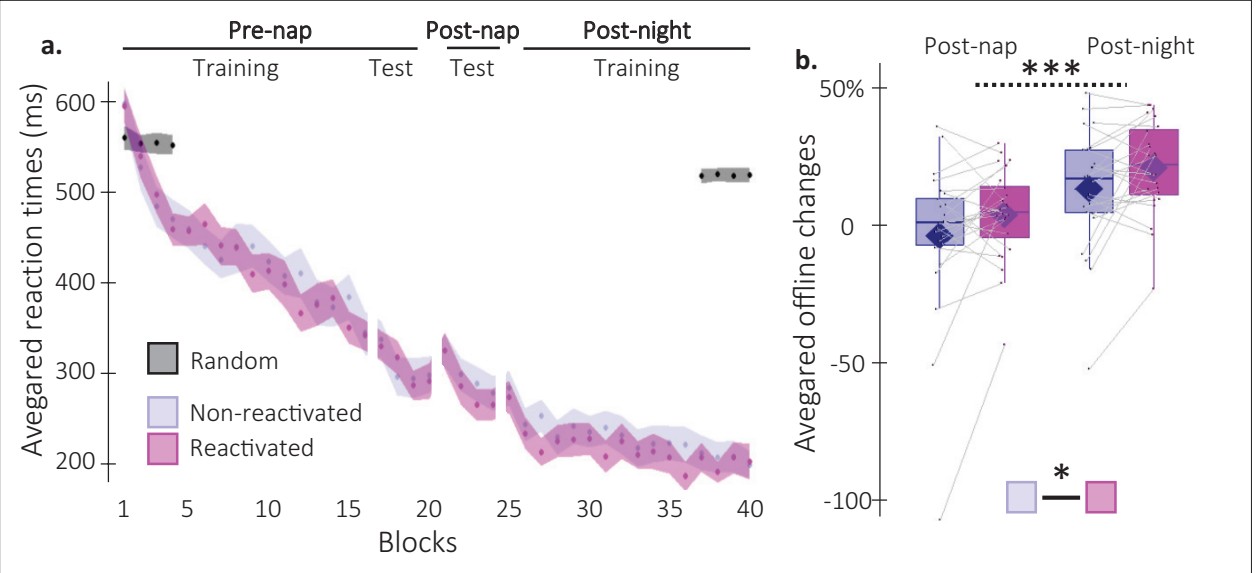

**Figure 2.** Behavioral results. (**a**) Performance speed (N = 24; mean reaction time in ms) across participants plotted as a function of blocks of practice during the pre- and post-nap sessions (+/-standard error in shaded regions) for the reactivated (magenta) and the non-reactivated (blue) sequences and for the random SRTT (Black overlay). (**b**) Offline changes **in performance speed** (N = 24; % of change) averaged across participants (box: median (horizontal bar), mean (diamond) and first(third) as lower(upper) limits; whiskers: 1.5 x InterQuartile Range (IQR)) for post-nap and post-night time-points and for reactivated (magenta) and non-reactivated (blue) sequences. Using a repeated-measure ANOVA, the results highlighted a main effect of Time-point (***: p-value <0.001) and a main effect of Condition (*: p-value <0.05). Note that the main effect of condition is marginally significant when excluding the extreme participant (p=0.077).

The online version of this article includes the following figure supplement(s) for figure 2:

**Figure supplement 1.** Behavioral results on performance accuracy.

**Figure supplement 2.** Behavioral results per sequence.

Block effect (F(2,46) = 1.56; p-value = 0.22). Similar to above, the main effect of Condition (F(1,23) = 0.04; p-value = 0.85) and the Block by Condition interaction (F(2,46) = 1.81; p-value = 0.18) were not significant. Altogether, these results indicate that a performance plateau was ultimately reached and both sequence conditions were learned similarly (*Figure 2a*).

Post-nap and post-night offline changes in performance were then computed for both conditions as the relative change in speed between the three plateau blocks of the pre-nap test and the first four blocks of the post-nap and post-night sessions, respectively. As such, improvement in performance from training to retest (i.e. faster performance at retest compared to training) was reflected by positive offline changes in performance. A repeated measures analysis of variance (rmANOVA) performed on offline changes in performance with Time-point (post-nap vs. post-night) and Condition (reactivated vs. non-reactivated) as within-subject factors showed a significant Time-point effect, whereby changes in performance were significantly higher at the post-night as compared to the post-nap retest (F(1,23) = 46.53; p-value = 5.89e-7; $\eta$ ²=0.67; *Figure 2b*). Critically, offline changes in performance for the reactivated sequence were significantly higher than for the non-reactivated sequence (Condition effect: F(1,23) = 4.75; p-value = 0.0397; $\eta$ ²=0.17). The Condition by Time-point interaction was not significant (F(1,23) = 7.42e-4; p-value = 0.98). In conclusion, our behavioral results indicate a TMR-induced enhancement in performance that did not differ across nap and night intervals.

## Electrophysiological data

Participants' sleep was recorded using a 6-channel EEG montage during a 90-min episode following learning. Sleep was monitored online and sounds were presented during NREM sleep stages. Sleep characteristics resulting from the offline sleep scoring as well as the distribution of auditory cues across sleep stages are shown in in *Supplementary file 1*. Briefly, results indicate that all the participants slept during the nap (average total sleep time: 67 min; average sleep efficiency: 74.9%) and that cues were accurately presented in NREM sleep (average stimulation accuracy: 88.4%).

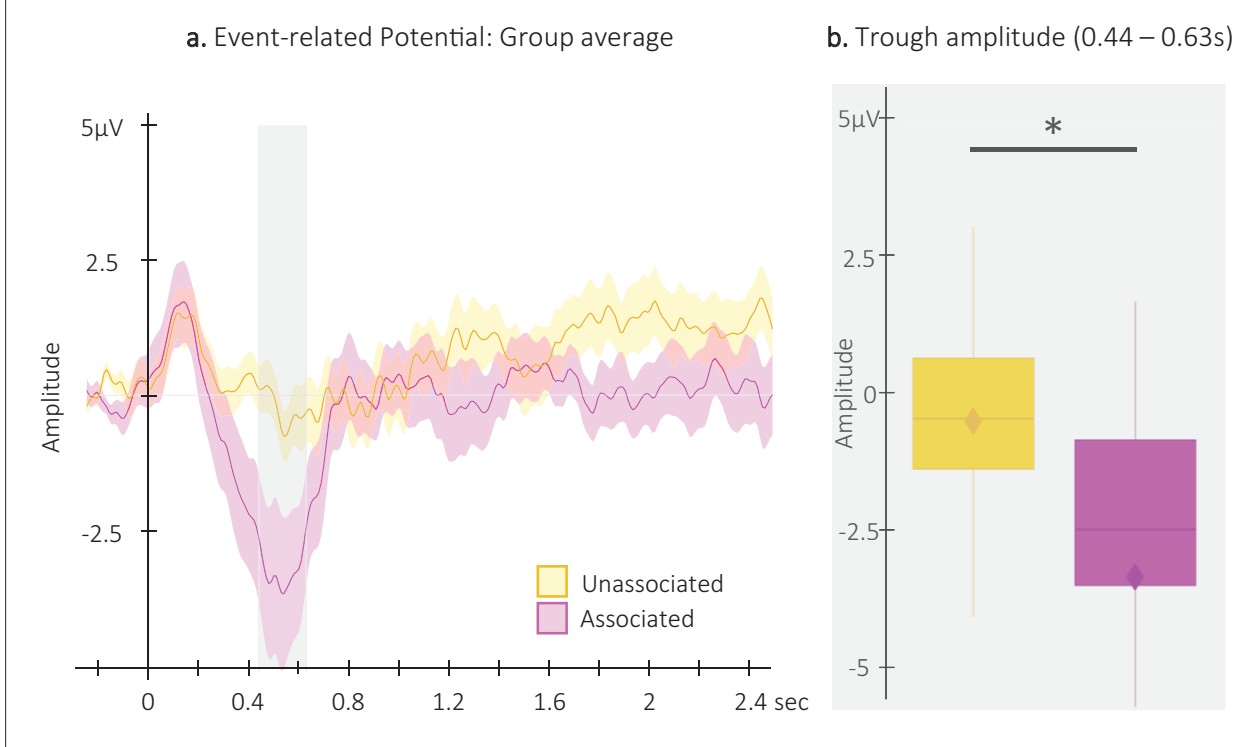

**Figure 3.** Event-related potentials (ERP). (**a**) Potentials averaged across all EEG channels and all participants (N = 24; +/-standard error in shaded regions) evoked by the associated (magenta) and the unassociated (yellow) auditory cues from –0.3 to 2.5 s relative cue onset. The gray region represents the temporal window (trough) in which ERPs across conditions were significantly different from zero. (**b**) ERP amplitude (N = 24; box: median (horizontal bar), mean (diamond) and first(third) as lower(upper) limits; whiskers: 1.5 x IQR) extracted from the temporal window highlighted in panel a, that is at 0.44 – 0.63 s post-cue onset (trough) in each condition. *: p-value <0.05 (Wilcoxon signed-rank test).

The online version of this article includes the following figure supplement(s) for figure 3:

**Figure supplement 1.** Topography of event-related potentials.

**Figure supplement 2.** Event-related potential across conditions.

**Figure supplement 3.** Topography of event-related potentials.

### Event-related analyses

Event-related analyses assessed the effect of the sound condition (i.e. associated vs. unassociated) on both the *potentials* (ERP) but also the *oscillatory activity* (time-frequency analyses) evoked by the auditory cues.

For the analyses of the auditory evoked potentials, we first computed ERPs on each EEG channel (see *Figure 3—figure supplement 1* for channel level data) separately for associated and unassociated auditory cues presented during NREM2-3 stages and subsequently averaged ERPs across channels (*Figure 3a*). ERP amplitude was extracted for the 2 conditions from the temporal window highlighted in *Figure 3a* in which the amplitude of the auditory responses across conditions was significantly lower (trough) than zero (from 0.44 to 0.63 s relative to cue onset, see *Figure 3—figure supplement 2* and *Figure 3—figure supplement 3* for across- and within-channel level data, respectively). Between-condition comparisons using Wilcoxon signed-rank test showed that the amplitude of the ERP trough was significantly deeper (V=75, p-value = 0.016) following associated as compared to unassociated cues (*Figure 3b*).

For the analyses of the auditory evoked *oscillatory* activity, we investigated whether EEG sigma oscillation power (12–16 Hz) evoked by the auditory cues on each channel was modulated by the different stimulation conditions in the 2.5 s following the cue onset. Note that, for completeness, time-frequency analyses were performed on a wider frequency range (5–30 Hz) and that analyses outside the sigma band were considered as exploratory. Cluster Based Permutations (*Maris and Oostenveld,*

*2007*) (CBP) tests computed on power averaged across all channels did not highlight any significant clusters between the two auditory cues.

## Sleep event detection

Slow waves (SWs) and spindles were detected automatically (*Vallat, 2020*) on all EEG channels in all NREM2-3 sleep epochs (thus including associated and unassociated sound stimulation intervals as well as non-stimulation intervals, see *Figure 1b*). The detection tool identified on average 424.8 [95% CI 328–521.6] slow waves and 98 [95% CI: 82.8–113.2] spindles averaged across channels during the nap episode (see methods for details on the detection algorithms and in *Supplementary file 2* for the number of events detected on each channel and each condition).

Concerning the detected SWs (*Figure 4a*), both peak-to-peak (PTP) amplitude and density (averaged across all EEG channels) were greater for the associated as compared to the unassociated stimulation intervals (amplitude: t=2.7; df = 21; p-value = 0.009; Cohen's d=0.55; and density V=197; p-value = 0.01; *r*=0.58). Exploratory analyses including the detected SWs in the non-stimulation (rest) intervals did not highlight PTP amplitude differences between the rest intervals and the two types of stimulation intervals (rest vs. associated: t=0.82; df = 21; p-value = 0.42; rest vs. unassociated: t=–0.92; df = 21; p-value = 0.42; *Figure 4b–c*). However, SW density was significantly lower during the rest as compared to the stimulation intervals, regardless of the cue type (rest vs. associated: V=232; p-uncorrected=0.0002; p-FDR=0.0004; *r*=0.66; rest vs. unassociated: V=224; p- uncorrected = 0.0008; p-FDR=0.00081; *r*=0.6; *Figure 4d*). Altogether, these results indicate that auditory stimulation induced an overall increase in SW density, and, more importantly, that the associated sounds resulted in an increase in SW amplitude and density as compared to the unassociated sounds.

Sleep spindle density averaged across all channels did not differ between associated and unassociated stimulation intervals (V=98, p-value = 0.89). Similarly, exploratory analyses on additional spindle features including amplitude and frequency did not yield any significant differences between stimulation conditions (all p-values > 0.2). As no effect of stimulation was observed on spindle characteristics, the two conditions were pooled together in exploratory analyses including spindles detected during rest intervals (*Figure 5*). Results show that spindle density did not differ between stimulation and rest intervals (V=97, p-value = 0.22). Interestingly, the difference in spindle amplitude was marginally significant with higher amplitudes during the auditory stimulation intervals as compared to the rest intervals (V=199; p-value = 0.065; *r*=0.73), whereas spindle frequency showed the opposite pattern (t=–3.42; df = 22; p-value = 0.005; Cohen's d=0.71). In summary, these results indicate that while auditory stimulation altered spindle features (frequency and amplitude to a lesser extent) as compared to rest, the two sound conditions did not differently influence these characteristics.

## Phase-amplitude coupling

We investigated whether the phase of the slow oscillations in the 0.5–2 Hz frequency band was coupled to the amplitude of sigma (12–16 Hz) oscillations following either the auditory cue or the negative peak of the detected (i.e. spontaneous) SWs. The analyses presented below focus on the comparison between conditions but see *Figure 6—figure supplement 1* for coupling analyses performed within each stimulation condition and at rest.

Event-related phase-amplitude coupling (ERPAC) analyses were performed across channels on a wider frequency range (7–30 Hz) for completeness; thus, analyses outside the pre-registered sigma band (see red frame in *Figure 6*) are considered exploratory. The ERPAC values locked to the *auditory cues* were compared between the two stimulation conditions. The CBP test did not highlight any significant clusters (alpha threshold = 0.025, cluster p-value = 0.44). The preferred coupling phase, which represents the phase at which the maximum amplitude is observed, did not significantly differ between conditions (F(1,46) = 0.3, p-value = 0.9). These results suggest that the stimulation conditions did not influence the coupling between the phase of the slow oscillations and the amplitude of sigma oscillations at the auditory cue.

Comparison of the ERPAC locked to the *negative peak of the SWs* (*Figure 6* and *Figure 6—figure supplement 2* for channel level data) between stimulation conditions revealed a significant cluster (alpha threshold = 0.025, cluster p-value = 0.024; Cohen's d=–0.56). Specifically, the coupling between the phase of the signal in the 0.5–2 Hz frequency band and the amplitude of the signal in the 14–18 Hz frequency band was significantly stronger around the negative SW peak (from –0.8 to

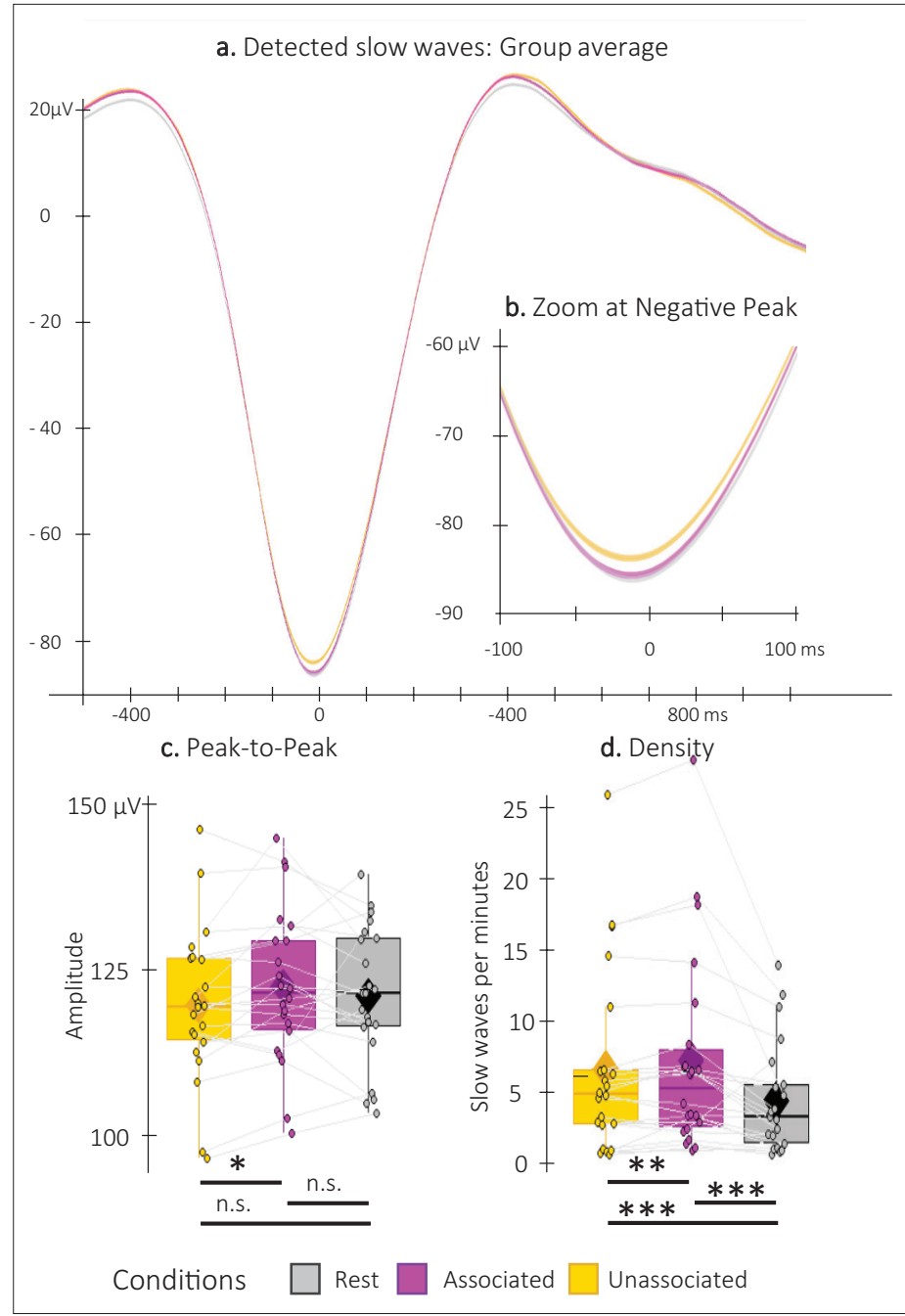

**Figure 4.** Detected Slow Waves (SWs). (**a**) Average at the negative peak (N = 22; +/-standard error) across all detected slow waves on all channels during the associated (magenta) and unassociated (yellow) stimulation intervals as well as in the rest (i.e. unstimulated) intervals (gray). (**b**) Zoom on the negative peak of the detected SWs. Shaded regions represent SEM. (**c**) Peak-to-peak SW amplitude (µV) was higher for associated as compared to unassociated sounds (Student t-test). (**d**) SW density (number of SWs per total time in minutes spent in stimulation or rest intervals) was higher during stimulation as compared to rest intervals and for associated as compared to unassociated sounds (Wilcoxon signed-rank test). Box: median (horizontal bar), mean (diamond) and first(third) as lower(upper) limits; whiskers: 1.5 x IQR; *: p-value <0.05; **: p-value <0.01; ***: p-value <0.001; n.s.: not significant .

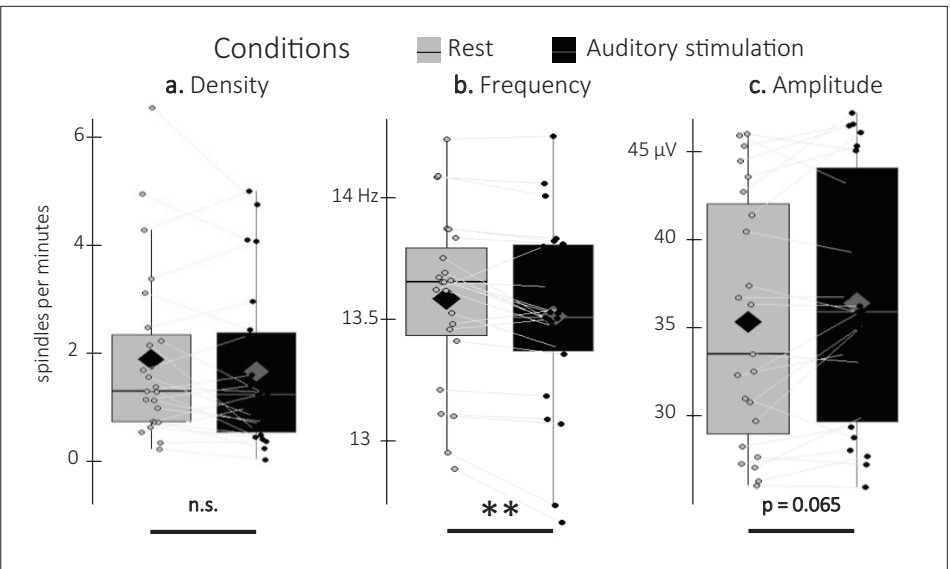

**Figure 5.** Detected spindles.
(**a**) Spindle density (number of spindles per total time in minute spent in stimulation or rest intervals) did not differ between stimulation intervals (irrespective of sound type; black) and rest (gray) intervals (Wilcoxon signed-rank test). (**b**) Spindle frequency (Hz) was lower during stimulation as compared to rest intervals (Student t-test). (**c**) Spindle amplitude (µV) was higher during stimulation as compared to rest intervals. All spindle features were averaged across channels (Wilcoxon signed-rank test). N = 23; Box: median (horizontal bar), mean (diamond) and first(third) as lower(upper) limits; whiskers: 1.5 x IQR; **: p-value <0.01; n.s.: not significant.

0.2 s relative to negative peak) during unassociated as compared to associated stimulation intervals (*Figure 6b*). The exploratory comparison between rest and associated stimulation intervals did not reveal any significant clusters (alpha threshold = 0.025, all cluster p-values > 0.6) but a significant cluster was observed between unassociated stimulation and rest (alpha threshold = 0.025, cluster p-value = 0.001; Cohen's d=0.53; *Figure 6c*). This cluster was observed between 13.5 and 20 Hz and –1–0.5 s around the negative peak of the SW. The preferred phases in each of the conditions were not significantly different (associated vs. unassociated: $F_{(1,42)} = 0;007$, p-value = 0.94; associated vs. rest: $F_{(1,42)} = 0.01$, p-value = 0.91; unassociated vs. rest: $F_{(1,42)} = 0.3$, p-value = 0.87; see *Figure 6—figure supplement 1*). Altogether, these results suggest that slowand sigma oscillation coupling observed just before the onset of the SW was stronger during unassociated as compared to associated and rest intervals but that the preferred coupling phase was not modulated by the experimental conditions.

## Correlational analyses

Correlation analyses between the TMR index (i.e. the difference in offline changes in performance – averaged across time points – between the reactivated and the non-reactivated sequence) and the density of SW and spindles as well as with the amplitude of the ERP did not yield any significant results (density of spontaneous SW: S=2486, p-value = 0.97) density of spontaneous spindles S=1412, p-value = 0.24; amplitude of the negative peak of the ERP S=2282, p-value = 0.73. However, the correlational CBP analysis between the TMR index and the difference in TF power elicited by the different auditory cues highlighted one significant cluster (alpha threshold = 0.025, cluster centered on 0.5 s post-cue p-value = 0.022, rho = - 0.46; *Figure 7a* and *Figure 7—figure supplement 1* for channel level data). For illustration purposes, we extracted the difference in TF power within the significant cluster included in the pre-registered frequency band (12–16 Hz) and from 0.35 to 1 s post-cue onset (see *Figure 3*). The resulting scatter plot presented in *Figure 7b* indicates that higher TMR index (i.e. greater behavioral benefit of TMR) was related to higher sigma oscillation power for the unassociated compared to the associated sound condition.

Finally, with respect to ERPAC-TMR index correlation analyses, no significant correlation was observed between the *auditory-locked* ERPAC metrics and the TMR index (alpha threshold = 0.025, cluster p-values >0.09). In contrast, cluster-based permutation correlational tests performed between

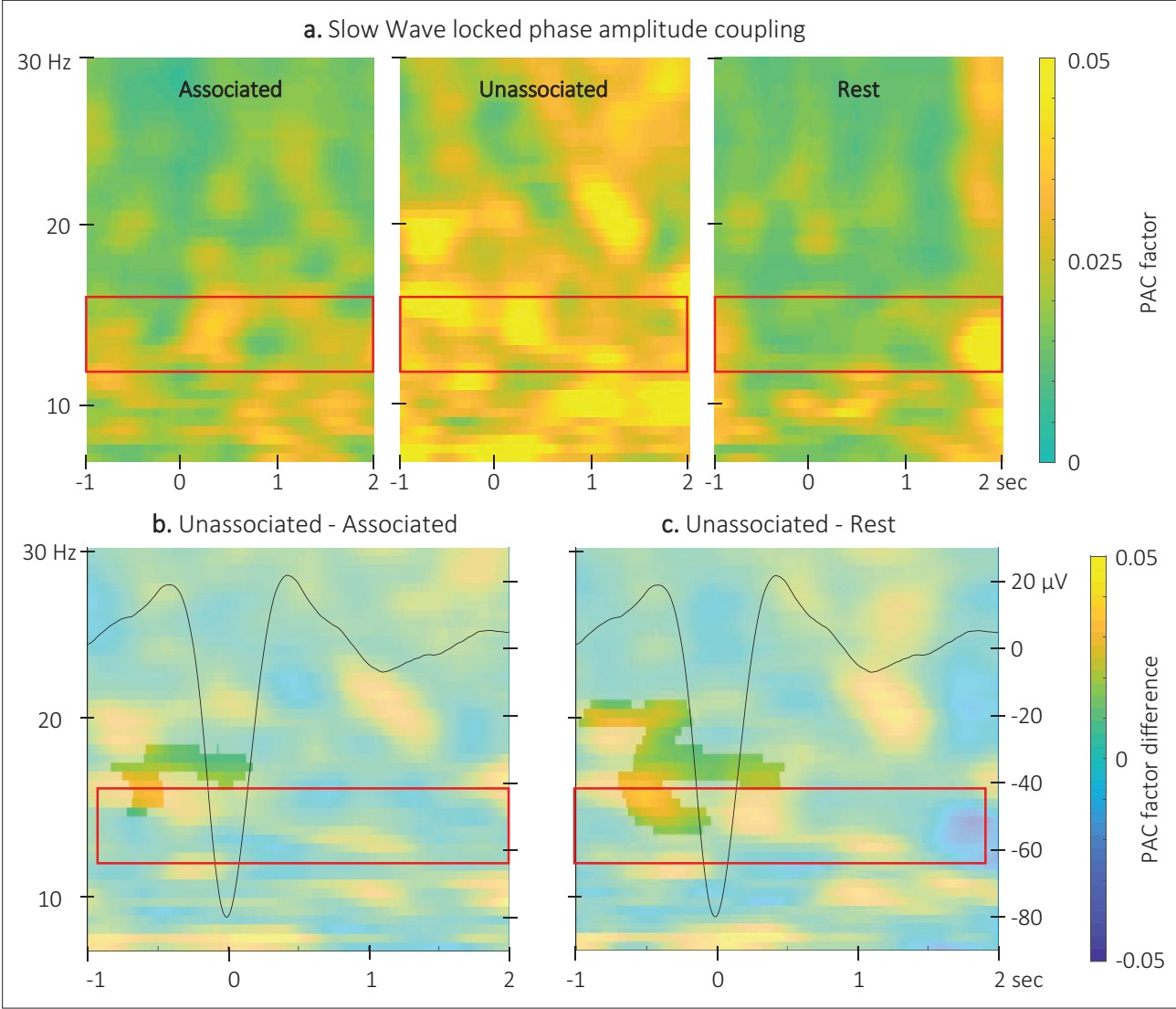

**Figure 6.** Event related phase-amplitude coupling locked to the detected slow wave negative peaks.
(**a**) Time-Frequency Representation (TFR) of group average (N = 22) coupling strength between the phase of the 0.5–2 Hz frequency band and the amplitude from 7 to 30 Hz (y-axis) from –1 to 2 s (x-axis) relative to SW negative peak for the three interval types. (**b**) ERPAC was significantly higher during the unassociated as compared to the associated sound intervals in the highlighted cluster (cluster-based permutation test). (**c**) ERPAC was significantly higher during the unassociated sound as compared to the rest intervals in the highlighted cluster (cluster-based permutation test). Red frames indicate the pre-registered sigma frequency band of interest. Superimposed on the TFR in panels b and c (black line): SW grand average across individuals and conditions (N = 22; y-axis on right).

The online version of this article includes the following figure supplement(s) for figure 6:

**Figure supplement 1.** Preferred phase.

**Figure supplement 2.** Topography of the phase-amplitude coupling locked to the detected slow wave negative peaks.

**Figure supplement 3.** Correlation between SW-locked phase-amplitude coupling during unassociated intervals and SW features (left panel: peak-to-peak amplitude, right panel: density).

the 12 and 16 Hz TFR *SW-locked* ERPAC difference between the two conditions and the TMR index revealed a significant cluster. Results show that the associated vs. unassociated difference in coupling strength between the phase of the signal in the 0.5–2 Hz frequency band and the amplitude of the signal in the 14.5–17 Hz frequency band, just after the SW peak (0.5 and 1 s), was positively correlated with the TMR index (alpha threshold = 0.025, cluster p-value = 0.0499, rho = 0.55; *Figure 8a* and *Figure 8—figure supplement 1* for channel level data). For illustration purposes, we extracted the difference in ERPAC in the significant cluster included in the pre-registered frequency band (between

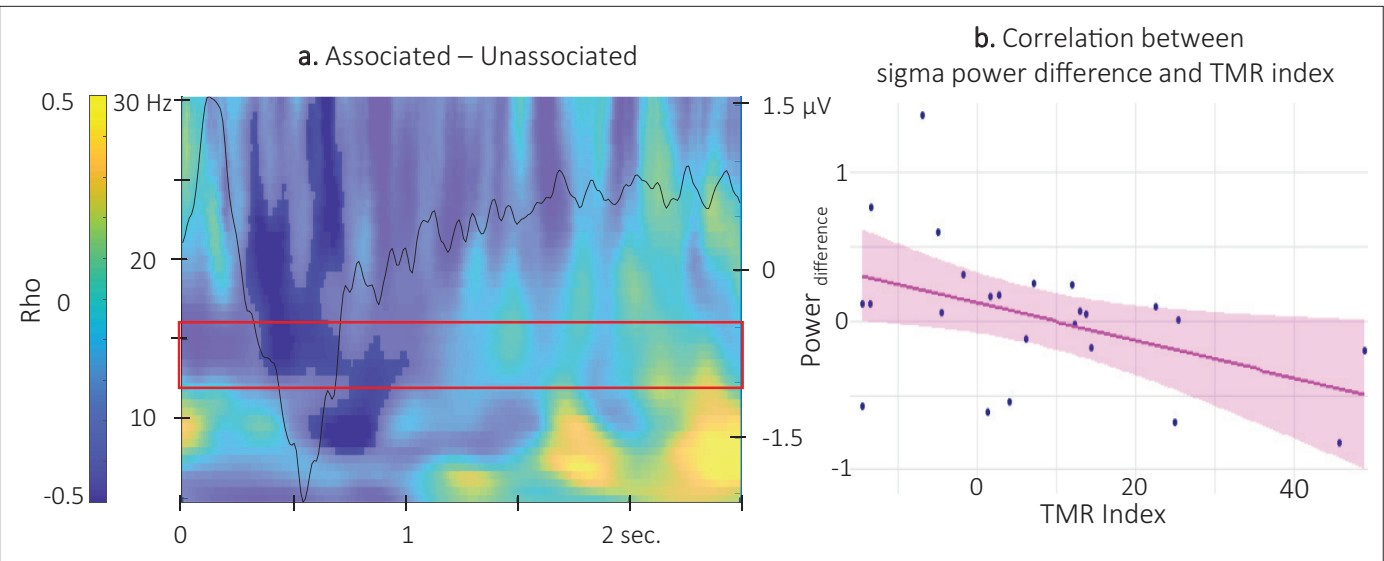

**Figure 7.** Correlation between power difference and TMR Index.
(a) Time-Frequency Representation (TFR) of the rho values issued from the correlation between the TMR index and the difference between the power elicited by the associated auditory cues and the unassociated ones (N = 24). Highlighted, the negative clusters in which the TMR index is significantly correlated with the difference in power (cluster-based permutation test). Red frame indicates the pre-registered sigma frequency band of interest. Superimposed on the TFR (black line): Grand average across individuals (N = 24) and conditions of event related potentials elicited by the auditory cues (y-axis on right). (b) Negative correlation between the difference in power elicited by the associated and unassociated cues (0.35–1 s post-cue, 12–16 Hz) and the TMR index (dots represent individual datapoints).

The online version of this article includes the following figure supplement(s) for figure 7:

**Figure supplement 1.** Topography of the correlation between power difference and TMR Index.

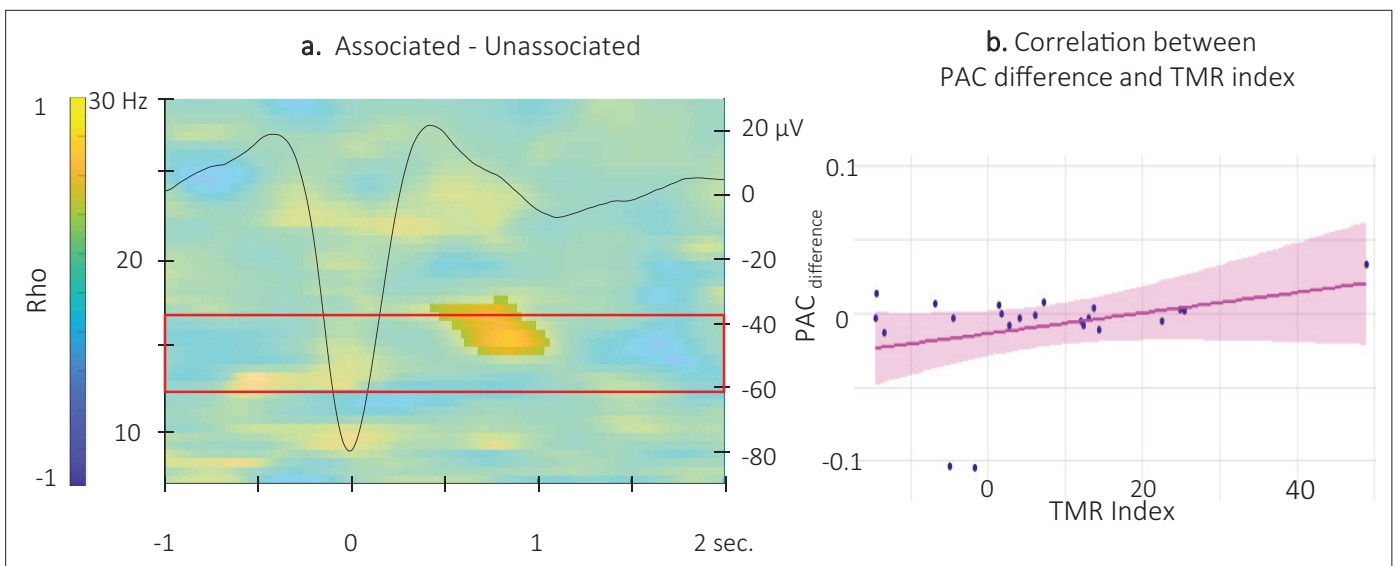

**Figure 8.** Correlation between SW-locked event related phase-amplitude coupling difference and TMR Index.
(a) Time-Frequency Representation (TFR) of the rho values issued from the correlation between the TMR index and the difference between the SW-locked ERPAC during the associated vs. unassociated stimulation intervals (N = 22). Highlighted, the positive cluster in which the TMR index is significantly correlated with the difference in SW-locked ERPAC (cluster-based permutation test). Superimposed on the TFR (black line): SW grand average across individuals and conditions. Red frame highlights the pre-registered sigma frequency band of interest. (b) Positive correlation between the SW-locked ERPAC difference (0.55–1.05 s post negative peak, 14.5–17 Hz) and the TMR index (dots represent individual datapoints).

The online version of this article includes the following figure supplement(s) for figure 8:

**Figure supplement 1.** Topography of the correlation between SW-locked phase-amplitude coupling difference and TMR Index.

14.5 and 16 Hz) from 0.55 to 1.05 s. The resulting scatter plot (*Figure 8b*) indicates that the stronger the phase-amplitude coupling during associated as compared to the unassociated stimulation intervals, the higher the TMR index.

## Discussion

In the present study, we examined the impact of auditory TMR on motor memory consolidation as well as the neurophysiological processes supporting reactivation during sleep. Our results demonstrate a TMR-induced behavioral advantage such that offline changes in performance were larger on the reactivated as compared to the non-reactivated sequence. These behavioral results are in line with earlier motor learning studies showing improvement in performance after auditory (*Schönauer et al., 2014*; *Antony et al., 2012*; *Cousins et al., 2016*) or olfactory (*Laventure et al., 2016*) TMR during sleep. As opposed to earlier TMR research though, the current results suggest that TMR-induced consolidation is not a protracted process that needs additional time and/or sleep to develop (*Cairney et al., 2018*), as a behavioral advantage could already be observed immediately after the TMR episode. Also, in contrast to earlier work showing that TMR effects can be transient (*Cousins et al., 2016*), the current data indicate that the effect of TMR on motor performance was sustained overnight. It remains unclear whether these discrepancies are related to the nature of the task (e.g. declarative vs. motor), the sensory stimulus used for reactivation (words vs. sound) or the duration of the reactivation / sleeping episode (nap vs. night). Nevertheless, our findings suggest that the TMR episode during a nap immediately following learning set the reactivated memory trace on a distinct yet parallel trajectory as compared to the non-reactivated memory trace.

TMR effects were also observed at the brain level such that electrophysiological responses differed according to whether they were evoked by associated or unassociated cues. Specifically, the amplitude of the negative component of the auditory ERP was higher for the sounds associated to the motor memory task as compared to the unassociated sounds. These results are in line with findings from earlier associative learning studies performed during wakefulness showing that auditory cues evoke larger responses after conditioning (i.e. after they are associated to another stimulus) and that ERP amplitude is restored to pre-association levels after extinction (see *Christoffersen and Schachtman, 2016* for a review). The current findings also extend prior observation of a modulation of auditory-TMR-evoked responses during sleep (*Rudoy et al., 2009*). This earlier study showed that auditory cues presented during post-learning sleep evoked larger ERPs when they were associated to items better remembered at subsequent recall as compared to cues associated to less remembered items. Our findings not only concur with this post-hoc analysis, but also provide the first direct evidence of an ERP modulation based on the memory content of the cue during post-learning sleep. This difference in brain potentials during sleep might be seen as the neural signature of the plasticity processes that occurred during learning. Not exclusive to the previous speculation, such effects might also be attributed to the (re)processing of the memory trace during post-learning sleep. Importantly, one could argue that the difference in ERP amplitude observed in the present study might be due to familiarity effects, as the unassociated sound might have been perceived as novel as compared to the associated sound. We argue that this is unlikely as new or rare auditory stimuli usually present larger negative amplitudes as compared to old or frequent sounds during both sleep and wakefulness (e.g. FN 400 [*Rugg and Curran, 2007*; *Paller et al., 2012*]) for old/new paradigms during wake and mismatch negativity components for oddball paradigms during both wake and sleep (*Näätänen et al., 2007*; *Ruby et al., 2008*). Instead, we propose that the auditory evoked brain responses observed in the current study reflect the (re)processing of the motor memory trace that was encoded during initial learning. It is worth noting that the negative peak of the potential evoked by the unassociated cues was not only lower (i.e. less negative) than for the associated cues but was also sometimes even absent on some channels (see *Figure 3—figure supplement 1*). These observations are partly in line with earlier research (*Cairney et al., 2018*) but contradict other findings. For example, *Weigenand et al., 2016* as well as *Schreiner and Rasch, 2015* observed significant negative responses evoked by control sounds or by sounds that were associated to later forgotten items. These discrepancies remain unclear but might in part be explained by methodological differences that are known to influence the amplitude of ERPs (e.g. sleep cycle stimulated, electrode(s) examined, number of stimulation repetitions provided).

In addition to the modulation of auditory-evoked responses described above, the properties of the detected (spontaneous) SWs were influenced by sound presentation and sound condition. Specifically, SW density was higher during sound presentation as compared to rest and the density and amplitude of the SWs were greater during intervals of associated – as compared to unassociated – cue stimulation. The effect of sound presentation on SW characteristics is in line with previous work showing sound-related entrainment of SW trains and increase in SW amplitude during sleep (*Ngo et al., 2013a*; *Ngo et al., 2013b*; *Ngo et al., 2015*; *Ngo et al., 2019*). More importantly, in line with the ERP results, our data show that the memory content of the cue modulated SW physiology above and beyond the mere effect of sound presentation. This is the first evidence, to the best of our knowledge, of a modulation of SW physiology based on the relevance of the sensory cues presented during sleep. We speculate that the processing of the memory content associated to the cue resulted in enhanced SW activity. Specifically, the greater amplitude of the SWs during associated sound intervals might reflect neural synchronization (*Carrier et al., 2011*; *Esser et al., 2007*) known to promote sleep-dependent plasticity processes for example (*Born and Wilhelm, 2012*; *Ngo et al., 2013b*). We thus propose that the TMR-effect observed in this study might therefore be mediated by SWs likely in relation with spindle activity.

While the characteristics of spontaneous spindles (amplitude and frequency) were only modulated by sound *presentation* and not sound *condition*, the properties of sigma oscillations (i.e. its amplitude and its coupling with the SO phase) were differently affected by the cue type and related to the TMR-induced behavioral advantage. The observation of a modulation of spindle characteristics irrespective of the sound condition suggests that spindle activity (in term of events) during reactivation is not related to motor memory processing per se. This stands in contrast with earlier reports of spindle-mediated effect of TMR on the consolidation of both declarative (*Cairney et al., 2018*) and motor (*Laventure et al., 2016*; *Cousins et al., 2016*) memory tasks. It is worth noting, however, that this earlier work did not compare different stimulus conditions as in the present study. Nonetheless, this previous research demonstrated that reactivation was related to an increase in spindle features (amplitude and frequency) that were linked to the TMR-induced motor performance advantage (*Laventure et al., 2016*).

Importantly, it is worth explicitly stating that our results do not rule out the involvement of spindle *activity* in TMR-related motor memory consolidation processes. Recent evidence has brought forward the idea that spindle event detection in general is less sensitive than the study of the sigma rhythm as a whole (*Dimitrov et al., 2021*). In line with these observations, our results show that sigma oscillation properties – as opposed to spindle events – were modulated by the sound condition and that such modulation was related to the TMR-induced behavioral advantage. Specifically, higher coupling between sigma oscillation amplitude and the SO phase, for associated as compared to unassociated sounds, on the descending phase following the peak of the SW was correlated with the TMR index. To the best of our knowledge, this is the first time that the strength of the coupling between the SO phase and the amplitude of the sigma oscillations nested within the peak of the SW is directly related to a TMR-related behavioral advantage. Earlier studies comparing different age groups provided convincing, yet indirect, evidence that the precise temporal coordination of SO and sleep spindles represents a critical mechanism for sleep-dependent memory consolidation (*Muehlroth et al., 2019*; *Helfrich et al., 2018*). The timing reported in this earlier work is consistent with the current data showing increased SW-sigma coupling on the descending phase following the peak of the SW. Our results are also in line with previous frameworks proposing that sigma oscillation (*Cairney et al., 2018*) / spindles (*Antony et al., 2019*) offer a privileged time window for relevant memories to be reinstated during sleep. Together with evidence that TMR boosted SW features, the current data suggest that both SWs and sigma oscillations play a critical role in the reactivation of motor memories.

In addition to the modulation of neurophysiological responses described above and triggered by the *associated* sounds, we report an intriguing pattern of brain results for the *unassociated* sounds. Specifically, the coupling between sigma amplitude and the SO phase was strengthened for unassociated sounds just *before the onset of the SW negative peak*. Furthermore, we observed that the increase in sigma power nested in the trough of the auditory evoked potential for unassociated (as compared to associated) sounds was related to higher TMR-induced performance enhancement. It is tempting to speculate that sigma oscillations might prevent the processing of unassociated/irrelevant sounds during post-learning sleep which might in turn be reflected by a decrease in the amplitude

of the slow electrophysiological responses (i.e. smaller ERP and SWs) during non-associated sound intervals. We argue that sigma oscillations might play the role of a gatekeeper for the consolidation process and protect the motor memory trace against potential interfering effects induced by the unassociated sounds which might in turn potentiate the effect of TMR at the behavioral level. In order to further examine this possibility, we performed additional exploratory analyses testing for potential relationships between the SW-sigma coupling observed during unassociated stimulation intervals and slow electrophysiological responses (see *Figure 6—figure supplement 3*). Results showed a negative correlation between SW-sigma coupling and SW features such that higher coupling was related to lower SW amplitude and density during unassociated stimulation intervals. These results provide further support for the protective effect of sigma oscillations (nested in the trough of slow oscillations) against potential interfering effects induced by the unassociated sounds. These assumptions are also in line with a growing body of literature pointing towards a sensory gating role of spindle activity / sigma oscillations (*Dang-Vu et al., 2011*; *Schabus et al., 2012*) that might be critical to facilitate the memory consolidation process during sleep (*Schreiner and Staudigl, 2020*; *Antony et al., 2019*). Specifically, it has been proposed that a function of the thalamus is to suppress distraction and gate information processing via alpha/beta oscillations during wakefulness (*Jensen and Mazaheri, 2010*) and sigma oscillations during sleep (*Chen et al., 2015*). Further support for the gating hypothesis comes from observations of both increased arousal threshold as well as decreased amplitude of auditory ERP when sounds are presented simultaneously to a spindle event (*Yamadori, 1971*; *Cote et al., 2000*). Along the same lines, previous studies using simultaneous EEG-fMRI recordings showed that the BOLD responses in relation to sound processing are inconsistent or even absent when sounds occur during sleep spindles or before the negative peak of the SW (*Dang-Vu et al., 2011*; *Schabus et al., 2012*). The present data therefore suggest that the precise SO-sigma coordination does not only play a role in the reinstatement of relevant memories, but is also critical to prevent the processing of irrelevant information during post-learning sleep. These observations are remarkably in line with recent theoretical views putting forward the concept that temporally precise SO-spindle coupling might not convey only memory-specific information (*Helfrich et al., 2021*). It is argued that while synchronized states might trigger memory reactivation, the underlying neural activity might offer limited opportunities for information processing. Our data concur with this theory as they suggest that SO-sigma coupling, depending on its temporal characteristics, either prevents the processing of irrelevant information or supports memory reactivation during post-learning sleep.

In conclusion, our results depict a complex organization of the different physiological processes supporting motor memory consolidation during post-learning sleep. While associated sounds appeared to boost SW features and SO-sigma coupling at the peak of the SW, unassociated sounds predominantly modulated the properties of the sigma oscillations at the trough of the slow oscillation. Our findings suggest a dual role of sigma oscillations whereby, depending on their temporal coordination with SWs, they either protect memories against irrelevant material processing or promote the reactivation of relevant motor memories; two processes that critically contribute to the motor memory consolidation process.

## Materials and methods

This study was pre-registered in the Open Science Framework (https://osf.io/). Our pre-registration document outlined our hypotheses and intended analysis plan as well as the statistical models used to test our a priori hypotheses (available at https://osf.io/y48wq). Whenever an analysis presented in the current paper was not pre-registered, it is referred to as *exploratory*. Additionally, to increase transparency, any deviation from the pre-registration is marked with a (#) symbol and listed in *Supplementary file 3* together with a justification for the change.

### Participants

Young healthy volunteers were recruited by local advertisements to participate in the present study. Participants gave written informed consent before participating in this research protocol, approved by the local Ethics Committee (B322201525025) and conducted according to the declaration of *World Medical Association, 2013*. The participants received a monetary compensation for their time and effort. Inclusion criteria were: (1) left- or right-handed[#] (see point #1 of *Supplementary file 3*); (2) no

previous extensive training with a musical instrument or as a professional typist, (3) free of medical, neurological, psychological, or psychiatric conditions, including depression and anxiety as assessed by the Beck's Depression (*Beck et al., 1996*) and Anxiety (*Beck et al., 1988*) Inventories, (4) no indications of abnormal sleep, as assessed by the Pittsburgh Sleep Quality Index (*Buysse et al., 1989*); (5) not considered extreme morning or evening types, as quantified with the Horne & Ostberg chronotype questionnaire (*Horne and Ostberg, 1976*); and, (6) free of psychoactive or sleep-affecting medications. None of the participants were shift-workers or took trans-meridian trips in the 3 months prior to participation.

The sample size was determined with a power analysis performed through the G*Power software (*Faul et al., 2007*) and based on the paper of *Cousins et al., 2016* which reports, to our knowledge, the closest paradigm to the present one in the motor memory domain (see details in the pre-registration). Sample size was estimated to 24 participants. Thirty-four participants took part in the study to reach this estimated sample size after participant exclusion. As per our pre-registration, participants were

**Table 1.** Participant characteristics and sleep characteristics leading up to the experimental session and vigilance assessments at time of testing.

| N | 24 (12 females) |
|---|---|
| Age (yrs) | 21.9 ranging from 18 to 27 |
| Edinburgh Handedness (*Oldfield, 1971*) | 78.6 [57.1–100] |
| Epworth Sleepiness Scale (*Hoddes et al., 1972*) | 7 [5.9–8.1] |
| Beck Depression Scale (*Beck et al., 1996*) | 1.5 [0.9–2.2] |
| Beck Anxiety Scale (*Beck et al., 1988*) | 1.8 [1.1–2.4] |
| PSQI (*Buysse et al., 1989*) | 3 [2.2–3.8] |
| Chronoscore (CRQ)(51) | 48.8 [45.6–51.9] |
| Sleep duration[a] | |
| Mean across the 3 nights (minutes) | |
| Night 1 | 481.5 [461.5–501.5] |
| Night 2 | 488.2 [471.2–505.2] |
| Night 3 | 502 [482.1–521.8] |
| St. Mary's questionnaire on Night 3 quality | |
| Quality | 4.7 [4.3–5.1] |
| Duration (minutes) | 443.5 [423.3–463.8] |
| Psychomotor Vigilance Task[b] | |
| Pre-nap | 300.2ms [289.7–310.6] |
| Post-nap | 297.5ms [288.9–306] |
| Post-night | 294.7ms [285.1–304.2] |
| One-way rmANOVA results | $F_{(2,46)}=1.47$; *P*-value = 0.24 |
| Stanford sleepiness score | |
| Pre-nap Session | 2.4 [2.1–2.7] |
| Post-nap Session | 2.7 [2.3–3.1] |
| Post-night Session | 2.3 [1.9–2.7] |
| One-way rmANOVA results | $F_{(2,46)}=1.69$; *P*-value = 0.2 |

*Notes.* Values are means [lower and upper limit of the 95% Confidence Interval - CI]. PSQI = Pittsburgh Sleep Quality Index; CRQ = Circadian Rhythm Questionnaire. REM: Rapid Eye Movement. [a] Sleep duration was computed as the mean across the actigraphy data and the sleep diary for the three nights before the experimental day. [b] Median of reaction times computed across the 100 trials of each session.

excluded if their sleep duration during the experimental nap was insufficient to provide at least 50 stimulations per condition (after EEG data cleansing). This cut-off aimed at providing enough events to reach sufficient signal-to-noise ratio for electrophysiological analyses. Ten participants did not reach this criterion; accordingly, 24 participants (12 females) completed the experimental protocol and were included in the analyses (see participants' characteristics in *Table 1*).

## General design

This study employed a within-participant design (*Figure 1*). Participants were first invited, in the early afternoon, for a habituation nap during which they completed a 90-min nap monitored with polysomnography (PSG, see below for details). Approximately 1 week later, participants returned to complete the experimental protocol. Each participant followed a constant sleep/wake schedule (according to their own rhythm +/-1 hr) for the 3 days before the experiment. Compliance was assessed with sleep diaries and wrist actigraphy (ActiGraph wGT3X-BT, Pensacola, FL). Sleep quality and quantity for the night preceding the experimental visit was assessed with the St. Mary's sleep questionnaire (*Ellis et al., 1981*; see *Table 1* for results about sleep data before the experimental session). During the first experimental day, participants were trained on two motor sequences simultaneously (pre-nap session: between 12 pm-1:30 pm). During learning, each of these two sequences was associated to a particular sound. Only one of these two sounds was presented during the subsequent nap episode, corresponding to the *associated* sound linked to the *reactivated* sequence. At the behavioral level, the control condition consisted of the *non-reactivated* sequence (i.e. a sequence that was associated to a sound during learning but the sound was not presented during the subsequent nap interval). For electrophysiological analyses, a new, *unassociated*, sound (i.e. a sound to which participants were not exposed during the learning episode) was presented during the post-learning sleep, serving as a control condition. The nap occurred between 1:30 pm and 3 pm and was monitored with PSG. Sleep data were monitored online by an experimenter in order to send auditory stimulations during NREM2-3 stages. Performance on the reactivated and non-reactivated sequences was tested 30 min after the end of the nap to allow sleep inertia to dissipate (post-nap session: 2 pm-5:30 pm) and on day 2 after a night of sleep (not monitored with PSG) spent at home (post-night session: 8:30 am-11:30 am). At the beginning of each behavioral session, vigilance was measured objectively and subjectively using the Psychomotor Vigilance Task (*Dinges and Powell, 1985*) and Stanford Sleepiness Scale (*Hoddes et al., 1972*), respectively (see *Table 1*). Finally, general motor execution was tested at the beginning of the pre-nap session and at the end of the post-night session.

This design allowed to assess the specific impact of TMR on consolidation at the behavioral level, with the comparison between the changes in performance of the reactivated and non-reactivated sequences assessed during the post-nap and post-night sessions; and at the electrophysiological level, with the comparison between the neurophysiological responses to the reactivated associated sound vs. the unassociated sound that did not carry mnemonic information.

## Stimuli and tasks

All tasks were performed on a laptop computer (Dell Latitude 5,490 run under Microsoft Windows 10 Enterprise) and were implemented in Matlab (Math Works Inc, Natick, MA, USA) Psychophysics Toolbox version 3 (*Kleiner, 2007*). Participants sat comfortably in front of the computer screen with the keyboard on their knees. This configuration allowed the participants to focus their gaze on the screen and not to look at their hands/movements. Distance between participants and the screen was approximately 70 cm but was self-selected by the participants based on comfort. The sound presentation was conducted using ER3C air tube insert earphones (Etymotic Research).

### Acoustic stimulation

Three different 100 ms sounds were randomly assigned to the three conditions (reactivated/associated, not-reactivated, and unassociated), for each participant. The three synthesized sounds consisted of a tonal harmonic complex created by summing a sinusoidal wave with a fundamental frequency of 543 Hz and 11 harmonics with linearly decreasing amplitude (i.e. the amplitude of successive harmonics is multiplied by values spaced evenly between 1 and 0.1); white noise band-passed between 100 and 1000 Hz and a tonal harmonic complex created with a fundamental frequency of 1480 Hz and 11 harmonics with linearly increasing amplitude (i.e. the amplitude of successive harmonics is multiplied

by values spaced evenly between 0.1 and 1). A 10 ms linear ramp was applied to the onset and offset of the sound files so as to avoid earphone clicks. At the start of the experiment, auditory detection thresholds were determined by the participants themselves using a transformed 1-down 1-up procedure (**Levitt, 1971**; **Leek, 2001**) separately for each of the three sounds. Subsequently, the sound pressure level was set to 1000% of the individual auditory threshold during the tasks and 140% for auditory stimulation during sleep, thus limiting the risk of awakening during the nap (**Sterpenich et al., 2014**). Before the start of the nap episode, participants were instructed that they may or may not receive auditory stimulations during the nap.

## Motor task

A bimanual serial reaction time task (**Nissen and Bullemer, 1987**) (SRTT) was used to probe motor learning and memory consolidation processes. During this task, eight squares were horizontally presented on the screen meridian, each corresponding to one of the eight keys on the specialized keyboard and to one of the 8 fingers (no thumbs). The color of the outline of the squares alternated between red and green, indicating rest and practice blocks, respectively. During the practice blocks, participants had to press as quickly as possible the key corresponding to the location of a green filled square that appeared on the screen. After a response, the next square changed to green with a response-to-stimulus interval of 0 ms. After 64 presses, the practice block automatically turned into a rest block and the outline of the squares changed from green to red. The rest interval was 15 s.

The order in which the squares were filled green (and thus the order of the key presses) either followed a sequential or pseudo-random pattern. In the sequential SRTT, that is assessing motor sequence learning, participants were trained simultaneously on two different eight-element sequences (sequence A: 1 6 3 5 4 8 2 7; sequence B: 7 2 6 4 5 1 8 3, in which 1 through 8 are the left pinky to the right pinky fingers respectively). Participants were explicitly told that the stimuli (and thus the finger presses) would follow two different repeating patterns composed of eight elements each, but were not told any further information. During each practice block, four repetitions of a specific sequence (e.g. sequence A) were performed, each separated by a 1 sec-interval. Then, after a 2 sec-interval, the four repetitions of the other sequence started (e.g. sequence B). The order of the two sequences was randomized within each block of practice. Each motor sequence was associated to a different tone that consisted of a single 100 ms auditory cue (see above). The auditory cue was presented before the beginning of each sequence repetition, that is before the first key press of the sequence that was to be performed. Accordingly, one single tone was associated to an eight-element sequence of finger movements. Participants were instructed to learn the sequence-sound association during task practice. The associations between sound-sequence (sounds 1, 2 and 3; sequence A, sequence B, and control sound presented during nap) and sequence-condition (sequences A and B; conditions reactivated and not-reactivated) were randomized, thus creating 12 different possible combinations of randomized variables. Each participant was pseudo-randomly assigned to one of these combinations, such that there were two participants per combination. For the random SRTT, the order of the eight keys was shuffled for each eight-element repetition and thus the number of each key press was constant across all random and sequential blocks. For both variants of the task, the participants were instructed to focus on both speed and accuracy.

For the pre-nap session, participants first completed 4 blocks of the random SRTT to assess general motor execution. Participants subsequently completed the sequential SRTT, which consisted of 16 blocks of training followed by 4 blocks of post-training test taking place after a 5 min break. This allowed the assessment of end of training performance after the further dissipation of physical and mental fatigue (**Pan and Rickard, 2015**). For the post-nap session, only 4 blocks of the sequential SRTT were completed to avoid extensive task practice before the final overnight retest. For the post-night session, 16 blocks of the sequential SRTT were performed, followed by 4 blocks of the random SRTT.

Between the training and test runs as well as after the post-night session, participants completed a generation task that aimed at testing explicit knowledge of the sequences as well as the strength of the association between the sequences and their corresponding auditory cues. During the generation task, participants were presented with the auditory cues specific to the learned sequences and were instructed to self-generate the corresponding motor sequences. Participants completed 4 consecutive attempts for each cue / sequence pair. The order of the pairs was randomized. Accuracy was

emphasized during this task. A trial was classified as 'correct' if the key pressed by the participant was in the correct ordinal position with respect to the sequence acoustically cued. The percentage of correct ordinal positions was computed per sequence and per attempt. The generation accuracy per sequence was computed by averaging across attempts for each time point separately (pre-nap and post-night sessions). We tested whether generation accuracy of the reactivated sequence during the pre-nap generation task was correlated (Pearson's correlation) to the TMR index. Results showed that there was no significant correlation between generation accuracy and the TMR index ($r$=0.25, t=1.22, df = 22, p-value = 0.24).

## Polysomnography and targeted memory reactivation protocol

Both habituation and experimental naps were monitored with a digital sleep recorder (V-Amp, Brain Products, Gilching, Germany; bandwidth: DC to Nyquist frequency) and were digitized at a sampling rate of 1000 Hz (except for one participant (500 Hz) due to experimental error). Standard electroencephalographic (EEG) recordings were made from Fz, C3, Cz, C4, Pz, Oz, A1, and A2 according to the international 10–20 system (note that Fz, Pz and Oz were omitted during the habituation nap). A2 was used as the recording reference and A1 as a supplemental individual EEG channel. An electrode placed on the middle of the forehead was used as the recording ground. Bipolar vertical and horizontal eye movements (electrooculogram: EOG) were recorded from electrodes placed above and below the right eye and on the outer canthus of both eyes, respectively. Bipolar submental electromyogram (EMG) recordings were made from the chin. Electrical noise was filtered using a 50 Hz notch. Impedance was kept below 5 kΩ for all electrodes. During the experimental nap, PSG recordings were monitored by a researcher in order to detect NREM2-3 sleep based on the most recent sleep scoring guidelines from the American Academy of Sleep Medicine (*Berry, 2018*). To do so, PSG recordings were displayed online using 30-second-long epochs with EEG and EOG data filtered from 0.5 to 30 Hz and EMG data filtered between 20 and 200 Hz. When NREM2-3 sleep stages were reached, auditory cues were sent. The auditory stimulation was presented in a blocked design (*Figure 1B*). Namely, each type of auditory cue (associated or unassociated) was sent during 3-min-long stimulation intervals with an inter-stimulus interval of 5 s. The stimulation was stopped manually when the experimenter detected REM sleep, NREM1 or wakefulness. Intervals of stimulation for each sound were separated by a 1 min silent period (rest intervals).

## Analysis

Statistical tests were performed with the open-source software (*R Development Core Team, 2020*; *RStudio Team, 2020*) and considered significant for p<0.05. When necessary, corrections for multiple comparisons was conducted with the False Discovery Rate (*Benjamini and Hochberg, 1995*) (FDR) procedure within each family of hypothesis tests (see details for each analysis below). Greenhouse-Geisser corrections was applied in the event of the violation of sphericity. Wilcoxon signed-rank tests were used when the Shapiro-Wilk test indicated non-normal distribution[#] (see point #6 of *Supplementary file 3*). F, t and V (or W) statistics and corrected p-values were therefore reported for ANOVAs, Student and Wilcoxon tests, respectively. Effect sizes are reported for significant comparisons using *Cohen's d* for Student t-tests, *r* for Wilcoxon signed-rank test and $\eta^2$ for rmANOVAs using G*power (*Faul et al., 2007*). For correlation analyses, Spearman[#] test (see point #6 of *Supplementary file 3*) was used and S as well as corrected p-values were reported. Nonparametric CBP tests (*Maris and Oostenveld, 2007*) implemented in fieldtrip toolbox (*Oostenveld et al., 2011*) were used for high dimensional time and time-frequency data analyses (e.g. ERP, TF, and PAC analyses). CBP tests are composed of two subsequent tests. The first calculates paired t-tests (for contrast analyses) between conditions for each time points (or time-frequency points), which are then thresholded at a chosen p-value which sets the conservativeness of the test (reported as 'cluster threshold'). Significant clusters are defined as showing a continuum of significant time (or time-frequency) points. Subsequently, the procedure is repeated 500 times on shuffled data in which the condition assignment within each individual is permuted randomly. On each permutation, the maximum t-value is retained, yielding a distribution of 500 t-values (for contrast analyses). Finally, this distribution is used as a reference to determine whether the statistical value (t in case of contrast analyses) of each cluster, as calculated on the real assignment of the conditions, is likely to come from the same probability distribution (p-value

>0.05) or rather differs significantly from this random perturbation probability distribution (p-value <0.05). For CBP contrast analyses, *Cohen's d* is reported while rho is reported for CBP correlations.

## Behavior
### Preprocessing
*Motor performance* on both the random and sequential SRTT was measured in terms of speed (correct response time (RT) in ms) and accuracy (% correct responses) for each block of practice. Note that RTs from individual correct trials were excluded from the analyses if they were greater than 3 standard deviations above or below the participant's mean correct response time for that block (1.73% in total). Consistent with our pre-registration, our primary analyses were performed on speed.

The *offline changes in performance* on the sequential SRTT were computed as the relative change in speed between the pre-nap session (namely the 3 last blocks of practice[#], see results and point #2 of ***Supplementary file 3*** for details) and the post-nap session (4 blocks of practice) and the post-night session (4 first blocks of practice) separately for the reactivated and the non-reactivated sequences. A positive offline change in performance therefore reflects an increase of absolute performance from the pre-nap test to the post-nap or post-night tests. Additionally, we computed a *TMR index,* to be used in brain-behavior correlation analyses, which consisted of the difference in offline changes in performance - averaged across time points - between the reactivated and non-reactivated sequences. A positive TMR index reflects higher offline changes in performance for the reactivated as compared to the non-reactivated sequence.

### Statistical analyses
We first assessed whether performance (speed and accuracy) significantly differs between conditions during initial training. To do so, two two-way rmANOVAs with Condition (reactivated vs. non-reactivated) and Block (1st rmANOVA on the 16 blocks of the pre-nap training and 2nd rmANOVA on the 4 blocks of the pre-nap test) as within-subject factors were performed on the sequential SSRT performance. Similar analyses testing for baseline differences between sequences A and B irrespective of the reactivation condition were performed. The results of these control analyses are presented in ***Figure 2—figure supplement 2***. We then tested whether offline changes in performance on the sequential SRTT differed between reactivation conditions after a nap and night of sleep. This was done with a rmANOVA with Time-point (post-nap vs. post-night) and Condition (reactivated vs. non-reactivated) as within-subject factors on the offline changes in performance. Finally, to highlight that improvement in movement speed was specific to the learned sequences as opposed to general improvement of motor execution, we computed the *overall performance change* for both the sequential SRTT (first 4 blocks of pre-nap raining vs. 4 last blocks of post-night training collapsed across reactivated and non-reactivated sequences) and the pseudo-random version of the SRTT (4 blocks pre-nap session vs. 4 blocks post-night session). Two-tailed paired Student t-test revealed that overall performance changes in performance were significantly higher for the sequential SRTT as compared to the random SRTT (t=21.69, df = 23, p-value <2.2e-16; Cohen's d=4.43). Thus, the RT decrease reported on the sequential SRTT in the result section reflect motor sequence learning rather than a mere improvement in motor execution.

## Electroencephalography
### Offline sleep scoring
A certified sleep technologist blind to the stimulation periods completed the sleep stage scoring offline according to criteria defined in ***Iber and Iber, 2007*** using the software SleepWorks (version 9.1.0 Build 3042, Natus Medical Incorporated, Ontario, Canada). Data were visually scored in 30 s epochs and band pass filters were applied between 0.3 and 35 Hz for EEG signals, 0.3 and 30 Hz for EOG, and 10 and 100 Hz for EMG. A 50 Hz notch filter was also used (see ***Table 1*** for details of extraction from scored data).

### Preprocessing
EEG data preprocessing was carried out using functions supplied by the fieldtrip toolbox (***Oostenveld et al., 2011***). Specifically, data were cleaned by manually screening each 30-s epoch. Data segments

contaminated with muscular activity or eye movements were excluded. Data were filtered between 0.1 and 30 Hz.

## Event-related analyses

Event-related data analyses (i.e. auditory-evoked potentials and oscillatory activity) were performed with the fieldtrip toolbox (*Oostenveld et al., 2011*) with down sampled data (100 Hz). Auditory-evoked responses were obtained by segmenting the data into epochs time-locked to auditory cue onset (from –1 to 3 s relative to auditory cue onset after correction for onset-trigger lags) separately for the associated and unassociated auditory cues and averaged across all trials[#] (see point #3 of *Supplementary file 3*) in each condition separately. During cleaning, 1.03% [95% CI: 0.49–1.58] of the trials with stimuli sent during NREM2-3 stages were discarded. The remaining number of artifact-free trials was not significantly different between the two stimulation conditions (associated vs. unassociated, t=–0.5888, df = 23, p-value = 0.5617).

For *event-related potentials (ERPs) analyses*, individual ERPs computed on each channel were baseline corrected by subtracting mean amplitude from –0.3 to –0.1 s relative to cue onset. As our low-density EEG montage did not allow to perform fine topographical analyses, ERP data were averaged across all 6 EEG channels (but see *Figure 3—figure supplement 1* and *Figure 3—figure supplement 3* where channel level data are presented) data. In a first step, we used CBP approaches on ERP data computed across conditions to identify specific time windows during which significant brain activity was evoked by the auditory stimulation (i.e. where ERPs were significantly greater than zero). Results showed that across condition ERP was significantly different from zero between 0.44 and 0.63 sec at the trough (alpha threshold = 0.025, cluster p-value = 0.044; Cohen's d=–0.56; see *Figure 3—figure supplement 2*). In a second step, ERP amplitude was then averaged within this specific time-window for the two conditions separately and compared using one-tailed paired Wilcoxon signed-rank test[#] (see point #6 of *Supplementary file 3*) with the hypothesis that ERP absolute amplitude at the trough is greater following the associated cues as compared to unassociated cues.

To analyze *oscillatory activity*, we computed Time-Frequency Representations (TFRs) of the power spectra per experimental condition and per channel. To this end, we used an adaptive sliding time window of five cycles length per frequency ($\Delta t=5$ /f; 20 ms step size), and estimated power using the Hanning taper/FFT approach between 5 and 30 Hz[#] (see point #4 of *Supplementary file 3*). Individual TFRs were converted into baseline relative change of power (baseline from –0.3 to –0.1 s relative to cue onset), thus highlighting power modulation following the auditory cues. All six EEG channels were then averaged (but see *Figure 7—figure supplement 1* for channel level data). To identify significant evoked power modulation, TFR locked to auditory cues were compared between conditions using a CBP test between 5 and 30 Hz and from 0 to 2.5 sec relative to cue onset.

## Sleep-event detection

Preprocessed cleaned data were down-sampled to 500 Hz and were transferred to the python environment. *S*low waves and spindles were detected automatically in NREM2-3 sleep epochs on all the channels, by using algorithms implemented in the YASA open-source Python toolbox (*Vallat, 2020*; *Vallat and Walker, 2021*). Concerning the SW detection, the algorithm used is a custom adaptation of *Massimini et al., 2004* and *Carrier et al., 2011*. Specifically, data were filtered between 0.3 and 2 Hz with a FIR filter using a 0.2 Hz transition resulting in a –6 dB points at 0.2 and 2.1 Hz. Then all the negative peaks with an amplitude between –40 and –200 µV and the positive peaks with an amplitude comprised between 10–150 µV are detected in the filtered signal. After sorting identified negative peaks with subsequent positive peaks, a set of logical thresholds are applied to identify the true slow waves: (1) duration of the negative peak between 0.3 and 1.5 sec; (2) duration of the positive peak between 0.1 and 1 sec; (3) amplitude of the negative peak between 40 and 300 µV; (4) amplitude of the positive peak between 10 and 200 µV and (5) PTP amplitude between 75 and 500 µV. Concerning spindle detection, the algorithm is inspired from the A7 algorithm described in *Lacourse et al., 2019*. Specifically, the relative power in the spindle frequency band (12–16 Hz) with respect to the total power in the broad-band frequency (1–30 Hz) is estimated based on Short-Time Fourier Transforms with 2-s windows and a 200ms overlap. Next, the algorithm uses a 300ms window with a step size of 100 ms to compute the moving root mean squared (RMS) of the filtered EEG data in the sigma band. A moving correlation between the broadband signal (1–30 Hz) and the EEG signal filtered in the spindle band is then computed. Sleep spindles are detected when the three following thresholds

are reached simultaneously:(1) the relative power in the sigma band (with respect to total power) is above 0.2 (2) the moving RMS crosses the $RMS_{mean}$ +1.5 $RMS_{SD}$ threshold and (3) the moving correlation described is above 0.65. Additionally, detected spindles shorter than 0.5 s or longer than 2 s were discarded. Spindles occurring in different channels within 500ms of each other were assumed to reflect the same spindle. In these cases, the spindles are merged together.

SWs and spindles were detected in the stimulation intervals of both associated and unassociated sounds. One participant did not show any SW during the unassociated cue stimulation intervals and the minimal required number of SWs was not reached to perform the PAC in another participant. The two participants were thus excluded from the analyses on detected SWs. No spindles were detected during the associated cue stimulation intervals for another participant who was therefore excluded from the spindle analyses. With respect to the detected SWs, we extracted for each participant, each condition and channel, the mean PTP amplitude (µV) of SWs[#] (see point #3 of *Supplementary file 3*) as well as their density (number of SWs per total time in minutes spent in stimulation or rest intervals). These characteristics were then averaged across channels. Concerning the spindles, we extracted for each participant, condition and channel, spindle density (i.e. the number of spindles per total time in minutes spent in stimulation or rest intervals). Spindle amplitude (computed as the PTP amplitude (µV) in the sigma-filtered data) and frequency were also extracted for exploratory analyses. These different dependent variables were then averaged across channels and were compared using a one-tailed paired Student t-test (SW PTP and spindle Frequency) or Wilcoxon signed-rank (SW density, spindle density and amplitude) test[#] (see point #6 of *Supplementary file 3*) with the hypothesis that the associated, as compared to unassociated, stimulation intervals would exhibit higher values.

Furthermore, we performed exploratory analyses including the SWs and the spindles detected during rest intervals (i.e. NREM 2–3 epochs without auditory stimulation). In the case of SWs, we compared these values with those obtained for the associated stimulation intervals and the unassociated stimulation intervals using two two-tailed Student t-tests or Wilcoxon signed-rank tests (rest vs. associated stimulation intervals and rest vs. unassociated stimulation intervals). In the case of spindles, as spindle characteristics did not differ between stimulation conditions (see results), they were collapsed across stimulation conditions and compared to rest intervals using two-tailed Student t-tests or Wilcoxon signed-rank tests. Correction for multiple comparisons was performed using the FDR approach (*Benjamini and Hochberg, 1995*).

## Phase-amplitude coupling

In order to perform coupling analyses, preprocessed cleaned data were first down-sampled to 500 Hz. Based on the low spatial resolution of our montage that did not allow fine topographical analyses and in order to increase the signal to noise ratio to enhance the quality of the phase estimation that is particularly sensitive to noise (*Gross et al., 2013*), we opted to average the data across channels (but see *Figure 6—figure supplement 2* and *Figure 8—figure supplement 1* for channel level data). Coupling analyses were then performed using the Event-Related Phase-Amplitude Coupling (ERPAC) method proposed by *Voytek et al., 2013* and implemented in the TensorPac to support multi-dimensional arrays (*Combrisson et al., 2020*). This method allows to compute the ERPAC at each time point of the analysis window (*Lachaux et al., 1999*) and is therefore optimal to preserve the time dimension. Specifically, the instantaneous phases of the slow oscillation (0.5–2 Hz) and the envelopes of amplitudes of the signal between 7 and 30 Hz[#] (see point # 4 of *Supplementary file 3*) were first calculated by Hilbert transform around the trials of interest (i.e. from –0.5 to 2.5 s around the auditory cue onset and from –1 to 2 s around the negative peak of the SWs). For each time point in the analysis window (i.e. every 2 ms), the circular-linear correlation of phase and amplitude values were computed across trials. This analysis therefore tested whether trial-by-trial differences in *slow oscillation* phase explained a significant amount of the inter-trial variability in signal amplitude in the analyzed time window. The PAC factor output therefore represents the corresponding correlation coefficient. ERPAC was computed separately for the two sound conditions and compared using CBP test. Additionally, we performed exploratory analyses in which ERPAC (computed relative to the negative peak of the SWs as described above) was extracted from rest intervals. We compared rest ERPAC to ERPAC derived from both the associated and unassociated stimulation intervals using CBP procedures and corrected for two comparisons using the FDR. The preferred phase (PP), which reflects whether the amplitude of the signal in a given frequency band is modulated by the phase of the signal in another band, was also computed using tensorPac (*Combrisson et al., 2020*) open-source Python

toolbox. Based on our a priori hypotheses, these analyses focused on the amplitude of the signal in the sigma band and the phase of the SO. The amplitude was binned according to phase slices. The preferred phase is given by the phase bin at which the amplitude is maximum. The PP statistical analyses were performed using the CircStat toolbox (*Berens, 2009*) implementing Rayleigh test for non-uniformity and Watson-Williams multi-sample test for equal means[#] (see point #5 of *Supplementary file 3*). Similar as above, PP was also extracted from rest intervals for exploratory analyses in which rest PP was compared to the PP derived from the two different stimulation intervals using Watson test for circular data.

## Correlational analyses

Following our pre-registration, we performed correlation analyses between the TMR index and the following EEG-derived data: (1) The difference between the densities of SWs detected during the associated and unassociated cue stimulation intervals using one-sided Spearman[#] correlations (point #6 of *Supplementary file 3*); (2) The difference between the densities of spindles detected during the associated and unassociated cue stimulation intervals using one-sided Spearman[#] correlations ; (3) The relative change between the amplitude of the negative peak of the ERP[#] (point #3 of *Supplementary file 3-3*) following the associated and unassociated auditory cues using one-sided Spearman[#] correlations ; (4) The difference in auditory-locked sigma band power (0–2.5 sec relative to cue onset and from 12 to 16 Hz) between the associated and unassociated auditory cues using CBP tests[#] (point #7 of *Supplementary file 3*); and (5) The difference between SO phase and sigma oscillation amplitude (12–16 Hz) coupling strength during the associated and unassociated stimulation intervals in relation to the cue onset and to the SW event using CBP approaches[#] (point #6 of *Supplementary file 3*). For all one-sided tests, we predicted that the TMR index would be positively correlated with the EEG-derived metrics.

# Acknowledgements

This work was supported by the Belgian Research Foundation Flanders (FWO; G0D7918N), The Fond de Recherche en santé du Québec en sciences naturelles (RRQNT-2018–264146), Healthy Brain for Healthy Lives Discovery Grant Program from the Canada First Research Excellence Fund and internal funds from KU Leuven. GA also received support from FWO (G0B1419N, G099516N, 1524218 N) and Excellence of Science (EOS, 30446199, MEMODYN, with SS). Financial support for authors JN and BRK was provided by the European Union's Horizon 2020 research and innovation program under the Marie Skłodowska-Curie grant agreement (#887,955 and #703490, respectively). Finally, we would like to thank dr. Raphaël Vallat for his valuable help when author JN first approached YASA open-source Python toolbox.

# Additional information

## Funding

| Funder | Grant reference number | Author |
|---|---|---|
| Fonds Wetenschappelijk Onderzoek | G0D7918N | Judith Nicolas<br>Bradley R King<br>David Levesque<br>Latifa Lazzouni<br>Stephan Swinnen<br>Julien Doyon<br>Julie Carrier<br>Genevieve Albouy |

| Funder | Grant reference number | Author |
|---|---|---|
| Fonds de Recherche du Québec - Santé | RRQNT-2018-264146 | Judith Nicolas<br>Bradley R King<br>David Levesque<br>Latifa Lazzouni<br>Stephan Swinnen<br>Julien Doyon<br>Julie Carrier<br>Genevieve Albouy |
| Fonds Wetenschappelijk Onderzoek | G0B1419N | Genevieve Albouy |
| Fonds Wetenschappelijk Onderzoek | G099516N | Genevieve Albouy |
| Fonds Wetenschappelijk Onderzoek | 1524218N | Genevieve Albouy |
| Fonds Wetenschappelijk Onderzoek | 30446199 | Stephan Swinnen<br>Genevieve Albouy |
| HORIZON EUROPE Marie Sklodowska-Curie Actions | 887955 | Bradley R King |
| HORIZON EUROPE Marie Sklodowska-Curie Actions | 703490 | Judith Nicolas |
| Healthy Brain for Healthy Lives Discovery Grant Program from the Canada First Research Excellence Fund | | Julien Doyon<br>Genevieve Albouy<br>Bradley R King<br>Julie Carrier<br>Emily Coffey |
| KU Leuven | | Genevieve Albouy |

The funders had no role in study design, data collection and interpretation, or the decision to submit the work for publication.

### Author contributions

Judith Nicolas, Methodology, Validation, Visualization, Data curation, Formal analysis, Funding acquisition, Project administration, Supervision, Writing - original draft, Writing - review and editing; Bradley R King, Methodology, Conceptualization, Data curation, Formal analysis, Investigation, Project administration, Resources, Software, Supervision, Writing - review and editing; David Levesque, Project administration, Software, Supervision; Latifa Lazzouni, Formal analysis, Software; Emily Coffey, Formal analysis, Software, Supervision; Stephan Swinnen, Methodology, Conceptualization, Software; Julien Doyon, Julie Carrier, Methodology, Conceptualization, Formal analysis, Investigation, Software; Genevieve Albouy, Methodology, Conceptualization, Data curation, Formal analysis, Funding acquisition, Investigation, Project administration, Resources, Software, Supervision, Writing - review and editing

### Author ORCIDs

Judith Nicolas 
Bradley R King 
Stephan Swinnen 
Julien Doyon 
Julie Carrier 
Genevieve Albouy 

### Ethics

Young healthy volunteers were recruited by local advertisements to participate in the present study. Participants gave written informed consent before participating in this research protocol, approved by the local Ethics Committee (B322201525025) and conducted according to the declaration of Helsinki (2013). The participants received a monetary compensation for their time and effort.

### Decision letter and Author response

Decision letter https://doi.org/10.7554/eLife.73930.sa1
Author response https://doi.org/10.7554/eLife.73930.sa2

# Additional files

## Supplementary files
- Supplementary file 1. Sleep and stimulation characteristics (N=24).
- Supplementary file 2. Mean number [lower and upper limit of the 95% CI] of sleep events across participants detected at the single channel level per condition of blocks (either stimulation associated/unassociated or rest).
- Supplementary file 3. List of the deviations from the pre-registered analyses followed by their justification. These deviations are marked with a # in the main manuscript.
- Transparent reporting form

## Data availability
All data can be found at https://zenodo.org/record/6642860#.Yqol46hBzD5. The source code is available at https://github.com/judithnicolas/MotorMemory_OpenLoop_TMR, copy archived at swh:1:rev:1300ddefdec0c9980058d378fd06eeb8119971c4.

The following dataset was generated:

| Author(s) | Year | Dataset title | Dataset URL | Database and Identifier |
|---|---|---|---|---|
| Nicolas J, Albouy G, King BR | 2022 | Data set of manuscript entiteld Sigma Oscillations Protect or Reinstate Motor Memory Depending on their Temporal Coordination with Slow Waves | https://doi.org/10.5281/zenodo.6642860 | Zenodo, 10.5281/zenodo.6642860 |

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
