## [Editor Report]

The authors demonstrate that targeted memory reactivation (TMR) can enhance motor memory consolidation. TMR has mainly been used to strengthen declarative memories, and, on a neurophysiological level, TMR has been shown to strengthen oscillatory cross-frequency coupling. Here the authors extend previous findings into the motor domain and reveal that TMR strengthens motor memories and again, cross-frequency coupling of cardinal sleep oscillations, namely slow waves and spindles. Collectively, their findings provide additional evidence for the idea that the temporal precision of cross-frequency network coordination is critical for timed information transfer from short-term to long-term mnemonic storage.

---

## [Decision Letter]

**Decision letter after peer review:**

Thank you for submitting your article "Sigma Oscillations Protect or Reinstate Motor Memory Depending on their Temporal Coordination with Slow Waves" for consideration by *eLife*. Your article has been reviewed by 3 peer reviewers, and the evaluation has been overseen by a Reviewing Editor and Chris Baker as the Senior Editor. The following individual involved in review of your submission has agreed to reveal their identity: Hong-Viet Ngo (Reviewer #3).

Essential revisions:

All three reviews highlight that (i) the study is well designed and with preregistration it has the necessary rigor, and (ii) that TMR in the context of motor memory consolidation constitutes an important finding for the field.

All three reviews agree that the methods lack detail and need to be improved and substantially expanded. The one key finding that needs to be better demonstrated is whether there is actually any meaningful coupling in this data set. As outlined below, all three reviews identified several issues with the employed methods. In general, visualizations could be improved to illustrate the findings that were described in the text.

*Reviewer #1 (Recommendations for the authors):*

– First of all, it was not possible to download the pre-registration on OSF mentioned in the manuscript.

– Figure 2: It is unclear whether the gains are indeed significantly different from zero in all conditions. In my understanding, this would be a prerequisite to talk about gains in the first place (This is especially important with regard to the conclusion: 'In conclusion, our behavioral results indicate a TMR-induced enhancement in performance that did not differ across nap and night intervals.'). In addition, the graphic is hard to read with regard to the within-person nature of the task, as between-person indicators of errors are shown (IQR). Would it be possible to change the figure in a way that (a) a single person data is shown and (b) the dots are connected in a way that they reflect the within-person change in gains from nap to over-night (separately for the two conditions)?

– ERP analyses: The authors report that they first tested the averaged ERP across all trials against zero with a cluster-based permutation approach. However, in the methods section, when describing the cluster-based permutation approach, they only refer to paired t-tests. But testing against zero basically refers to a one-sample t-test against zero. How exactly was the cluster-based permutation approach implemented in this case?

In addition, it is not obvious why there was a need to follow this two-step approach in the first place. Would it not be possible to directly test for the condition-effect (unassociated vs. associated cues) directly with the cluster-based permutation framework?

By and large, the latter approach would also allow testing for topographic differences, the authors just mention with respect to a figure in the supplemental materials.

– Table 2 does not specify the unit of the depicted values.

– Figure 5 implies a mean amplitude for spindle events of around ~ 60 µV. This would be fairly large and in the range of typical slow-wave amplitudes.

– From the description of the PAC analyses on page 21, it is unclear how exactly the procedure was implemented. Specifically, it is described that, “all 6 EEG channels were averaged together”. If this was done prior to PAC analyses, one would run the danger that the original phase-relationships get distorted via the averaging.

Also, it is unclear whether analyses were performed on averages or on single trial data.

In my opinion, the preferred analysis pipeline would start with calculating PAC on the level of single trials and single channels. Averaging should then be performed for whole PAC matrices first across trials, within channels and only in the end across channels.

– Concerning the PAC analyses, it remains unclear whether there is a reliable SO-SP coupling in the first place. The comparison of PAC values across conditions is not very telling as long as there is no significant coupling to begin with. Again, here the authors would first need to test whether there is PAC significantly higher than chance / zero, before comparing conditions.

*Reviewer #2 (Recommendations for the authors):*

(1) Calculation of offline gains.

Post-nap as well as post-night offline gains in performance were calculated as a relative change to pre-nap performance. I’m wondering whether it would make sense to calculate the post-night offline gains relative to the post-nap performance (and not the pre-nap performance). Therefore, a direct comparison between post-nap and post-night offline gains is possible (does the night after TMR results in a significantly higher gain than a nap+TMR?).

(2) Mismatch in data points?

In the method section it says (page 18): “For the post-nap session, only 4 blocks of the sequential SRTT…” In Figure 2a the post-nap test shows data points of 5 blocks. Where does the 5th data point”come from?

(3) Figure 2b.

Showing the main effect of conditions like that is misleading. The reader could interpret it as the difference between non-reactivated and reactivated conditions being significant only for the post-nap offline gains.

(4) Showing single subject data and the individual change between conditions.

– Figure 4 c and d. This is just a suggestion but I think showing individual data points (and the single subject change across the three conditions) would improve understanding the highly significant effects. For example, in Figure 4d the density of SWs for the associated and unassociated condition almost looks the same. However, as this statistical comparison is a within comparison (and relies on the change within a participant) showing the single subject change in addition to the averages per condition increases the readability of the statistical finding.

– Figure 2b. Seeing data points and density plots would be helpful especially given the probably skewed distribution for the post-nap offline gains (for the non-reactivated sequence).

– same applies to Figure 5.

(5) Heading titles.

I’m not entirely sure whether this is in the realm of the journal, but the Results section might benefit from headings shortly summarising the paragraph (e.g. on page 3. Instead of 2.1. behavioural data you could say something like 2.1. TMR enhanced behavioural performance)

(6) Figure 3.

Using the same colour scheme as in Figure 2 but for different conditions (violet for non-reactivated in Figure 2 and unassociated in Figure 3) is misleading.

(7) Table 1.

To increase the reading flow of the paper and solely focus on the main findings, the authors might want to put table 1 into the supplement. Table 1 contains relevant information to underpin that participants slept and that cues were mainly delivered in NREM sleep but it is not fundamental for the general research question.

(8) No negative peak for unassociated sounds.

It is very interesting that the unassociated sounds did not elicit any negative peak. Even though the authors mention this (for me) surprising result in the discussion, I am missing referencing the literature demonstrating an evoked response even to sounds which were not associated with any memory content before (e.g. Cairney et al. (2018), Weigenand(2017)) or which were associated with later forgotten memories (Schreiner et al. (2015)). How do the authors interpret their findings in light of this literature?

Cairney SA, Guttesen A á V, El Marj N, Staresina BP. (2018). Memory Consolidation Is Linked to Spindle– Mediated Information Processing during Sleep. Current Biology, 28(6). 948-954.e4.

Weigenand et al. (2017). Timing matters: open-loop stimulation does not improve overnight consolidation of word pairs in humans. European Journal of Neuroscience. 45, 629-630.

Schreiner, T., Lehmann, M. and Rasch, B. (2015). Auditory feedback blocks memory benefits of cueing during sleep. Nature Communications 6, 8729.

(9) Averaging across all channels.

Why do the authors average the data across channels?

The ERPs clearly differ across channels (what is explicitly stated in the supplement page 4/5: "The effect reported in the main manuscript across channels is more pronounced on the frontal and the central electrodes" and what is visible in Figure S3)? To argue the averaging in the Results section, they cite Cairney et al. (2018) but I think this is not a convincing justification as Cairney et al. (2018) focused on multivariate analyses where all channels are included in the analyses. However, this is not the case in this study.

Further, the TFRs were averaged across channels as well as the data for the phase amplitude coupling. It has been shown that the phase of SOs in which spindles are coupled to differ across the scalp (e.g. Klinzing et al. (2016)).

I'm wondering to what extent the channel average conceals for example a SW-spindle coupling during rest which is not visible in Figure 6a but can be expected. The results might be more robust if the analyses were focused on the central region (which can be argued as the task is a motor learning task)?

Klinzing, J.G., Mölle, M., Weber, F., Supp, G., Hipp, J.F., Engel, A.K. and Born, J. (2016). Spindle activity phase-locked to sleep slow oscillations. Neuroimage, 134 (2016), pp. 607-616.

(10) 2.2.1. Event-related analysis. Page 5: "Cluster based permutations tests did not highlight any clusters between the two auditory cues." Here, a CBP was conducted across time to compare unassociated and associated conditions, right? Maybe the authors want to report this result when they talk about the comparison between unassociated and associated conditions (more at the beginning of this paragraph) but this is just a suggestion.

(11) 2.2.2. Sleep event detection. Page 5 : "[…] see methods for details on the detection algorithms and Table S2 in Supplementary file 2". I believe the info is found in table S4.

(12) Detection algorithms.

Could the authors specify how exactly they detected SWs and spindles? I assume they band passed filtered the signal in the specific frequency range (0.5 – 2 Hz for SWs and 12-16Hz for spindles)? For spindles root mean square was used or was power extracted with the Hilbert transformation? The signal had to exceed a specific threshold for 0.5-3sec to be defined as a spindle? If so, what was the threshold?

(13) Page 6, first paragraph: "[…] more importantly, that the associated sounds resulted in an increase in SW amplitude, density and slope as compared to the unassociated sounds." Did the authors compare the slope?

(14) Differences in preferred phase between cue-locked and SW-locked

Maybe the authors can plot Table 2 as a rose plot? That way the results would be easier to read. I assume the preferred phase in Table 2 is indicated in radians?

Why do the authors see completely different phases of the coupling for the cued locked vs. the spontaneous SWs? Wouldn't you assume to have a similar coupling for evoked responses and spontaneous SWs with spindles?

(15) Specify how PAC is calculated.

In the method section it says: "The strength of the coupling between the phase of the SO signal and the amplitude of the 7-30Hz signal was then computed at each timepoint of the analysis window (every 2ms)." Please state more clearly how the PAC has been calculated and what the PAC factor is (Figure 6a). Did the authors consider using the mean vector length as a measure for the coupling strength?

(16) No PAC between SWs and spindles in rest condition.

I'm surprised that there seems to be no coupling between the SO phase and the sigma frequencies in the rest condition.

(17) Figure 6.

The color scheme is misleading as in both figures (6a and 6b) the same color scheme is used (ranging from blue to yellow) but the numeric range is different (Figure 6a starts at 0 and is just positively scaled and Figure 6b ranges from -0.025 to 0.05). The authors might want to adapt the color range in Figure 6a.

The c-axis (Figure 6b, PAC factor difference) is not symmetric.

(18) Figure 6a. unassociated condition.

It is noticeable that the PAC is especially high in many frequencies and at different timepoints for the unassociated condition. Of course, a statistical comparison is needed to test that (can be done by a CBP against 0). Do the authors have an explanation for that?

(19) 2.3. Correlation analyses.

Page 10.: "[…] Correlation analyses between the TMR index (i.e., the difference in offline gains)" Post-nap or post-night offline gains?

(20) 2.3. Correlation analyses.

Page 10: "[…] that higher TMR index was related to higher sigma oscillation power for the unassociated compared to the associated sound condition" Is it possible that the TMR benefit (higher TMR index) is driven by lower power for the associated cues (potentially due to a stronger evoked response and hence a stronger evoked down-state during which the spindle power in general is lower)?

(21) FDR correction in results.

The authors mentioned in the method section that p-values were corrected using FDR. It would be good to define in the result section whether a p-value was corrected or not.

(22) Figure 7. Could you please plot the correlation values between TMR index and TFR power differences as these are the values for the CBP if I understand correctly. Additionally, I'm interested in seeing the TFRs for the associated and unassociated cue locked responses (before taking the difference).

(23)Figure 8. Similar to Figure 7, please plot the correlation values as well.

*Reviewer #3 (Recommendations for the authors):*

– Abstract: The sentence beginning with "Importantly, sounds that were not associated…" is a bit complicated. Perhaps this can be rephrased?

– Behavioral results (page 3): Lower reaction times at the SRTT obviously reflect better performance. Hence, to me, it was puzzling at first to see that the interpretation of offline gains is flipped. It might help to add one short sentence to clarify this.

– TFR and PAC results (page 5, line 19 and page 8, line 12). Why were the TFR and PAC analysis performed on different lower limits (5 vs. 7 Hz).

– Page 6, line 20: To some spindle "features" and "characteristics" could mean the same. I would suggest to be more specific and phrase it as "spindle occurrence" or similar.

– Figure 3A and 4A/B: Is it possible to move the x-axis of Figure 3A to the bottom and add more labels to the ticks? Otherwise, it is difficult to interpret the timing. The same could be applied to Figure 4A. Moreover, the y-labels for Figure 4B could move a bit to the left.

– Color bars: If the color range for the TFR and PAC maps include negative values, I would like to suggest using a symmetric color bar. It would help the reader visualize where Zero lies.

– Figure 7A: The overlay of significant correlation cluster onto the power difference between associated – unassociated is very confusing. Why didn't the authors directly plot the correlation values? Combining this with a reporting of the TFR plots (see above) would give a much better overview of the important patterns.

– Phase-amplitude coupling (page 21): I do not understand how the PAC is computed. It is clear that the 0.5 to 2 Hz signal serves as the phase-reference and is related to power from 7 to 30 Hz in steps of 0.5 Hz. In my view, for each trial the phase of the peak in the related powerband is determined. Circular statistics can then determine if there is a non-uniform coupling and, if so, to which phase it corresponds. However, here the authors state that every 2 ms the strength of coupling is calculated. This means that at every time bin, frequency bin and trial, yields a pair of slow-wave phase and power value. How is the PAC derived from this data? More importantly, SO frequency may differ between subjects and trials, thus I cannot quite grasp how this can be generalized across subjects with a timewise x-axis. I hope the authors understand my confusion and I would appreciate it if they could elaborate on this analysis. Please note also, that a PAC analysis strongly depends on the present power, thus the comparison of the cue-locked conditions might be a confounded difference in slow wave power.

– I found it very interesting that a TMR benefit was found immediately after the nap, which was even improved further after an overnight. This contradicts recent evidence on episodic memory only showing a benefit after the additional overnight (Cairney et al., 2018). Furthermore, while this study uses an unfamiliar sound as a control condition, nap studies are ideal to implement a wake control group. It might be worthwhile to discuss or present as food for thought how the reported results related to episodic memory or what different studies design might bring as insights.

– In line with my last comment, it is interesting that the authors interpret their findings as opposing facilitating and protective mechanisms mediated by slow waves and sleep spindles, but what are the practical implications? On the one hand, stronger responses upon control cues are beneficial. On the other hand, the slow wave-spindle coupling plays an important role as well, however, in this case for real cues. Does this mean that TMR studies missed out on incorporating a control cue or is it enough to only cue with unfamiliar sounds to protect memories? Is something reactivated during control sounds? Given that control sounds don't really evoke a slow oscillatory response (whereas SW-locked analysis are performed across the while cueing interval) implies that control cue-related mechanisms might emerge after or between cueing.

A potential approach to understand all this is multivariate analysis see Cairney et al. 2018; Schreiner et al. 2020 or new preprints from the Lewis lab=. Thus, it might be worthwhile again to discuss the distinct functions of control and real cues with regard to memory reactivation.

[Editors' note: further revisions were suggested prior to acceptance, as described below.]

Thank you for resubmitting your work entitled "Sigma Oscillations Protect or Reinstate Motor Memory Depending on their Temporal Coordination with Slow Waves" for further consideration by *eLife*. Your revised article has been evaluated by Chris Baker (Senior Editor) and a Reviewing Editor.

The manuscript has been substantially improved and there are solely two methodological remaining issues that need to be addressed, as raised by Reviewer 2 and outlined below regarding the analysis of cross-frequency coupling in the data.

*Reviewer #2 (Recommendations for the authors):*

Thanks a lot for a very thorough revision.

The majority of my points have been addressed. However, there are still two of my comments (and the authors' responses) I have questions about:

Comment #4

Thanks for clarifying the phase-amplitude coupling analysis. However, it deviates from the pre-registration, doesn't it? In the pre-registration, the authors describe the coupling analysis as a phase-phase coupling analysis:

"SO-spindle coupling: Finally, we aim to determine preferred phase of SO-spindle coupling for both evoked and spontaneous oscillations. We will extract the instantaneous phase of both the SO-filtered signal and of the envelope power in the spindle frequency band. Then we will calculate the circular distance between the phase time series. The preferred phase result from the mean of the circular angle values and will be computed across all trials of each condition separately. "

I did not find any justification of that deviation in supplementary file 3. Why did the authors change their analysis approach? Do the results differ?

Comment # 20

Thanks a lot to the authors for all their effort to address my concern. However, I still have some concerns when comparing the main with the control analysis:

The authors used the procedure by Mikutta et al. (2019) to show that there is SW-sigma oscillation coupling in all three conditions (associated, unassociated, rest). The control analysis is completely valid and compelling to demonstrate SW-sigma oscillation coupling during the rest condition. However, when comparing the main analysis with the control analysis there seems to be some differences:

First, there is no difference in the preferred phase angle of the SW-filtered signal when the sigma oscillations peak. Wouldn't you expect a mean phase difference between the associated and unassociated condition given that the coupling is stronger around the SW trough in the unassociated condition (similarly to Figure 6 in the main text)?

Second, I would like to see the presence of SW-sigma oscillation coupling with their actual dependent variable (PAC factor) as this is the variable the authors based their findings on. They stated that a statistical comparison to 0 is not a suitable approach as the ERPAC ranges between 0 and 1. Consequently, significant differences between the data and 0 are very likely.

An alternative way to create control data (where you don't expect any coupling) is to use events without any SW/cue. For example, you can run the SW detection on your data. For each detected SW you can choose a control event which is a SW free event temporally close to the detected SW (e.g., within 30s pre or post). Based on these SW free events the same analysis as in Figure 6 can be run and statistical comparisons can be made. For a comparable approach see (Ngo et al., 2020).

References

Ngo, H. V. V., Fell, J., and Staresina, B. (2020). Sleep spindles mediate hippocampal-neocortical coupling during long-duration ripples. *ELife*, 9, 1-18. https://doi.org/10.7554/*eLife*.57011

---

## [Author Response]

Essential revisions:All three reviews highlight that (i) the study is well designed and with preregistration it has the necessary rigor, and (ii) that TMR in the context of motor memory consolidation constitutes an important finding for the field.All three reviews agree that the methods lack detail and need to be improved and substantially expanded. The one key finding that needs to be better demonstrated is whether there is actually any meaningful coupling in this data set. As outlined below, all three reviews identified several issues with the employed methods. In general, visualizations could be improved to illustrate the findings that were described in the text.

We appreciate the positive comments of the reviewers about our research.

We would also like to thank the reviewers for pointing out that the methods lacked details for some analyses. Accordingly, the methods section was significantly expanded, in particular the description of the algorithms employed in the detection of sleep events (see comment #16 of reviewer #2) and the methods related to the phase amplitude coupling analyses (see comment #6 of reviewer #1, comment #19 of reviewer #2 and comment #11 of reviewer #3).

We are also grateful to have the opportunity to clarify some of our methodological choices. All three reviewers requested more justification to be provided with respect to the averaging that was performed across EEG electrodes. This choice was made based on the low-density nature of our montage (6 channels), preventing us from performing meaningful topographical analyses. Averaging across channels instead improved signal-to-noise ratio. Accordingly, the channel level was not considered a factor of interest in the present study. Nevertheless, in order to address the reviewers’ comments (i.e., comments #3 and #6 of reviewer #1 as well as comments #3 and #12 of reviewer #2), we now provide channel level results in the supplemental material of the revised manuscript. Briefly, channel level data for ERP, TF and PAC analyses are consistent – but overall noisier – than averaged data reported in the main text.

In order to address the coupling point raised by two reviewers (comment #7 of reviewer #1 and comments #2 and #20 of reviewer #2), we now report additional analyses investigating coupling at rest and within each stimulation condition. Results show that there is indeed reliable coupling between slow oscillation phase and sigma band amplitude at rest and during auditory stimulation.

Last, with respect to the visualization of the data, all the figures of the manuscript have been thoroughly revised based on the very constructive comments and helpful suggestions of the reviewers.

We believe that the additional analyses requested by the reviewers as well as the recommended figure changes have greatly improved both the quality and the readability of the manuscript. We hope the editor and the reviewers find these changes satisfactory.

Reviewer #1 (Recommendations for the authors):– First of all, it was not possible to download the pre-registration on OSF mentioned in the manuscript.

We apologize for this error. We mistakenly provided the link to the private instead of the public repository. The public repository can be accessed following this link https://osf.io/y48wq that was updated in the revised manuscript.

– Figure 2: It is unclear whether the gains are indeed significantly different from zero in all conditions. In my understanding, this would be a prerequisite to talk about gains in the first place (This is especially important with regard to the conclusion: 'In conclusion, our behavioral results indicate a TMR-induced enhancement in performance that did not differ across nap and night intervals.'). In addition, the graphic is hard to read with regard to the within-person nature of the task, as between-person indicators of errors are shown (IQR). Would it be possible to change the figure in a way that (a) a single person data is shown and (b) the dots are connected in a way that they reflect the within-person change in gains from nap to over-night (separately for the two conditions)?

We do agree with the reviewer that the word “gains” suggests a positive change in performance that is significantly different from zero. Accordingly, we replaced all instances of “gain” by “change” in performance in the revised manuscript in order to alleviate the confusion. Note that the amplitude of offline changes in performance (i.e., whether or not the values are significantly different from zero) is heavily influenced by computational and/or methodological choices (e.g. number of blocks selected, presence/absence of immediate post-training test dissipating fatigue (Rickard et al., 2008) (Rickard and Pan, 2017)). Accordingly, it is recommended to *not* report comparisons to a test value of zero, but rather to compare offline changes between conditions and/or groups (King et al., 2017). In the context of the current study, behavioral results show that offline changes in performance were significantly greater for the reactivated – as compared to the non-reactivated – sequence, which suggests that TMR enhanced performance.

We thank the reviewer for the suggestion of adding individual data points to Figure 2b, and we changed it accordingly. Concerning the second suggestion, Author response image 1 reflects the within-person change in performance from post-nap to post-night tests (i.e., the time-point effect). However, we elected to keep the original layout in the revised manuscript, as it better illustrates the within-person contribution to the condition effect (i.e., reactivated vs. non-reactivated) that was of primary interest in the present research.

**Author response image 1. sa2fig1:** Offline changes in performance speed averaged across participants (box: median (horizontal bar), mean (diamond) and first(third) as lower(upper) limits; whiskers: 1). 5 x InterQuartile Range (IQR) for post-nap and post-night time-points and for reactivated (magenta) and non-reactivated (blue) sequences.

It is worth noting that the inclusion of individual data points in Figure 2b highlights one participant with extreme offline changes in performance. Importantly, this participant was not an outlier based on the pre-registered procedure and was therefore not excluded from the analyses. Nonetheless, we re-ran the rmANOVA without this participant. The condition effect is marginally significant when excluding this participant (F(1,22) = 3.4, p-value = 0.077). This information was added to the caption of Figure 2b in the revised manuscript for the sake of completeness.

– ERP analyses: The authors report that they first tested the averaged ERP across all trials against zero with a cluster-based permutation approach. However, in the methods section, when describing the cluster-based permutation approach, they only refer to paired t-tests. But testing against zero basically refers to a one-sample t-test against zero. How exactly was the cluster-based permutation approach implemented in this case?In addition, it is not obvious why there was a need to follow this two-step approach in the first place. Would it not be possible to directly test for the condition-effect (unassociated vs. associated cues) directly with the cluster-based permutation framework?By and large, the latter approach would also allow testing for topographic differences, the authors just mention with respect to a figure in the supplemental materials.

We thank the reviewer for pointing out this lack of clarity. We indeed performed a one-sample t-test against zero. This information was clarified in the revised manuscript (see below and p. 20, l. 29 in the revised manuscript).

It is indeed possible to test for condition effect using CBP. Yet, we elected to first identify temporal windows in which significant brain activity was evoked by the auditory stimulus (see comment #7 of this reviewer below in the context of the PAC analyses) before comparing conditions. It was indeed hypothesized that the different conditions would specifically alter the amplitude of brain responses during a significant ERP modulation (i.e., in a time-window where signal changes were above noise) which is why we adopted this two-step approach.

With respect to the comment on topography, it is also indeed possible to consider the channel level in the CBP analyses. We elected to not do so based on the low resolution of our montage (6 channels) and therefore averaged responses across channels. The approach we employed and as described above allowed us to reduce the dimensionality of the data (in both the temporal and spatial dimensions). Nevertheless, in order to address the reviewer’s comment, we performed the suggested analyses and directly compared the conditions per channel using CBP on the entire time-window. As expected, this more conservative approach showed no differences between conditions. However, and as shown in the Supplemental Figure S2 where ERPs are depicted per condition on each individual channel, the condition effect was more pronounced on central electrodes located above the motor cortex. CBP analyses testing for condition effects and performed on these central channels (i.e., averaged across Cz, C3, and C4, as per the suggestion of reviewer #2 in comment #13) revealed a significant cluster (p = 0.049) between 0.39 and 0.62 sec post-cue (i.e. in the same time window highlighted by our two-step approach).

We acknowledge that this two-step approach requires further justification in the revised manuscript. Corresponding changes can be found on:

p. 20, l. 27: “In a first step, we used CBP approaches on ERP data computed across conditions to identify specific time windows during which significant brain activity was evoked by the auditory stimulation (i.e., where ERPs were significantly greater than zero).”

The topography point raised by the reviewer is further addressed below in response to similar comments about channel averaging (comment #6 of this reviewer and comments #3 and #13 of reviewer 2). We invite the reviewer to consult these responses to evaluate the changes that were made to the manuscript to address this issue.

– Table 2 does not specify the unit of the depicted values.

We thank the reviewer for pointing out this omission. However, this table was removed in the revised manuscript and replaced by rose plots in response to comment #18 of reviewer #2 (see Supplemental Figure S5).

– Figure 5 implies a mean amplitude for spindle events of around ~ 60 µV. This would be fairly large and in the range of typical slow-wave amplitudes.

We agree with the reviewer that spindle amplitudes reported in the present paper are higher than what is usually observed in the literature. This discrepancy with previous reports is due to the (default) methods implemented in YASA to compute peak-to-peak spindle amplitude. Specifically, while the majority of the available detection algorithms compute spindle amplitude on filtered EEG data (Lacourse et al., 2019, Purcell et al., 2017), YASA extracts spindle peak-to-peak amplitude from the raw EEG signal by default. We agree with the reviewer that this might be confusing for the reader. Consequently, we now compute spindle amplitude on the filtered EEG data as in previous research. The mean amplitude reported in the revised version of the manuscript is now around 35 microvolts which is in line with previous work. Note that the main effect of sound on spindle amplitude reported in the initial manuscript is no longer significant with the new amplitude metrics (p = 0.065). The corresponding results have been updated in the revised manuscript (see Results section p. 6 l. 11 and Figure 5).

– From the description of the PAC analyses on page 21, it is unclear how exactly the procedure was implemented. Specifically, it is described that, “all 6 EEG channels were averaged together”. If this was done prior to PAC analyses, one would run the danger that the original phase-relationships get distorted via the averaging.Also, it is unclear whether analyses were performed on averages or on single trial data.In my opinion, the preferred analysis pipeline would start with calculating PAC on the level of single trials and single channels. Averaging should then be performed for whole PAC matrices first across trials, within channels and only in the end across channels.

We thank the reviewer for pointing out that the methods describing the PAC analyses were not detailed enough in the original manuscript. This section was significantly expanded in the revised manuscript (see p. 21, l. 47). Averaging across EEG channels was indeed performed at the raw signal level, i.e. prior to performing the phase-amplitude coupling (PAC) analysis. To explain this methodological choice, we would first like to clarify the PAC method that was employed in the present research. Coupling analyses were performed using the Event-Related Phase-Amplitude Coupling (ERPAC) method initially proposed by Voytek et al. (2013) – and later largely adopted in the field (e.g. Samaha et al., 2015, Arnal et al., 2015, Ladenbauer et al., 2017) – that is implemented in the TensorPac to support multi-dimensional arrays (Combrisson et al., 2020). Unlike blocked PAC that measures PAC across time cycle (e.g. with mean vector length algorithm computed on each and every trial), ERPAC is calculated at each time point (Lachaux et al., 1999) in order to preserve the time dimension. It determines the amount of trial-by-trial variance in the higher frequency sigma band amplitude that can be explained by trial-by-trial variations in slow oscillation phase and calculates the correlation between sigma amplitude and slow oscillation (or the regression between them) at each time point. This method allows to unravel PAC dynamic in response to a stimulus.

We decided to average the raw signal across channels prior to PAC analyses to increase the signal to noise ratio (SNR) as the quality of the phase estimation is particularly sensitive to noise (Gross et al., 2013). The reviewer is correct that, traditionally, averaging is performed across trials (such as for ERP and time-frequency power estimation after convolution). However, as mentioned above, such averaging strategy is not possible due to the nature of the ERPAC method used. Altogether, in order to increase SNR and based on the fact that (1) our montage did not allow fine topographical analyses and (2) averaging in the spatial domain is rather standard when computing global field power (Murray et al. 2008) that is also used for PAC estimation (Busch and Vanrullen, 2010), we opted to average raw signal at the channel level.

Nevertheless, in order to address the reviewer’s comment on channel level data (as well as similar comments raised in comment #13 of reviewer #2 who was also interested in seeing PAC results at the channel level), we now also report channel level data – as well as corresponding cluster-based permutation tests – for each PAC analyses reported in the main text (see Supplemental Figure S6 for PAC between conditions and Supplemental Figure S8 for PAC/TMR index correlation analyses in the revised submission). Briefly, channel level data revealed that, as expected, central – and to a lesser extent frontal – electrodes mainly contributed to the pattern of results highlighted on averaged maps reported in the main text.

We have also changed the main manuscript in order to clarify this pipeline:

p. 21, l. 47: “In order to perform coupling analyses, preprocessed cleaned data were first down-sampled to 500 Hz. Based on the low spatial resolution of our montage that did not allow fine topographical analyses and in order to increase the signal to noise ratio to enhance the quality of the phase estimation that is particularly sensitive to noise (73), we opted to average the data across channels (but see Supplemental Figures S6 and S8 for channel level data). Coupling analyses were then performed using the Event-Related Phase-Amplitude Coupling (ERPAC) method proposed by Voytek et al. (2013) and implemented in the TensorPac to support multi-dimensional arrays (71). This method allows to compute the ERPAC at each time point of the analysis window (72) and is therefore optimal to preserve the time dimension. Specifically, the instantaneous phases of the slow oscillation (0.5-2 Hz) and the envelopes of amplitudes of the signal between 7-30 Hz# (see Supplemental Table S3.4) were first calculated by Hilbert transform around the trials of interest (i.e, from -0.5 to 2.5 sec around the auditory cue onset and from -1 to 2 sec around the negative peak of the SWs). For each time point in the analysis window (i.e., every 2ms), the circular-linear correlation of phase and amplitude values were computed across trials. This analysis therefore tested whether trial-by-trial differences in slow oscillation phase explained a significant amount of the inter-trial variability in signal amplitude in the analyzed time window. The PAC factor output therefore represents the corresponding correlation coefficient. ERPAC was computed separately for the two sound conditions and compared using CBP test.”

– Concerning the PAC analyses, it remains unclear whether there is a reliable SO-SP coupling in the first place. The comparison of PAC values across conditions is not very telling as long as there is no significant coupling to begin with. Again, here the authors would first need to test whether there is PAC significantly higher than chance / zero, before comparing conditions.

We thank the reviewer for this suggestion. However, the precise analysis outlined by the reviewer would not be overly meaningful with the ERPAC method employed in this study. Notably, as ERPAC values range from 0 to 1, an ERPAC significantly different from zero is essentially a certainty. We therefore opted to follow the procedure used in Mikutta et al. (2019) in order test for the presence of coupling during rest and stimulation intervals. Specifically, we tested whether the amplitude of the sigma oscillations peaked at a preferred phase of the slow oscillation across trials within each stimulation condition and at rest. These analyses were performed for both cued- and SW-locked analyses (in 2.5 sec and 3 sec analysis time-windows, respectively, consistent with the analyses reported in the main text). Specifically, we tested whether the preferred phases were uniformly distributed using Rayleigh test for non-uniformity of circular data (Berens, 2009); with a non-uniform distribution of the preferred phase being an indicator of coupling. Results show that the cue-locked preferred phases across trials were not uniformly distributed in both sound conditions (associated: Rayleigh z = 10.6, p-value = 7.7e-6 (7.7e-6 FDR-corrected); unassociated cues: Rayleigh z = 15.8, p-value = 4e-9 (8.1e-9 FDR-corrected)). SW-locked analyses revealed that the phase at which the amplitude was the highest was also not distributed uniformly during associated (Rayleigh z = 9.7, p-value = 2e-5 (6e-5 FDR-corrected)), unassociated (Rayleigh z = 4.8, p-value = 0.007, (6.9e-3 FDR-corrected)) and rest (Rayleigh z = 5.7, p-value = 0.003 (3.8-3 FDR-corrected)) intervals. Altogether, these results indicate significant SW-sigma coupling within each stimulation condition and at rest. These results are now presented in the Supplemental Figure (S5) and changes have been made in the methods (p. 22, l. 22) and results (p. 8, l. 10, l. 16, and l. 29) sections accordingly.

Reviewer #2 (Recommendations for the authors):(1) Calculation of offline gains.Post-nap as well as post-night offline gains in performance were calculated as a relative change to pre-nap performance. I’m wondering whether it would make sense to calculate the post-night offline gains relative to the post-nap performance (and not the pre-nap performance). Therefore, a direct comparison between post-nap and post-night offline gains is possible (does the night after TMR results in a significantly higher gain than a nap+TMR?).

We thank the reviewer for this suggestion. However, the results coming from such computation might be difficult to interpret as data from the post-nap session (as opposed to the pre-nap session) are already influenced by the intervention and can therefore not be considered as baseline performance. We therefore opted to compute post-night offline changes in performance using baseline performance assessed during the pre-nap session as done in previous research (e.g. King et al., 2017; Rumpf et al., 2017; Albouy et al., 2016).

(2) Mismatch in data points?In the method section it says (page 18): “For the post-nap session, only 4 blocks of the sequential SRTT…” In Figure 2a the post-nap test shows data points of 5 blocks. Where does the 5th data point”come from?

We thank the reviewer for bringing this error to our attention. The post-nap session indeed only included 4 blocks. The first block of the retest session was erroneously plotted as 5^th^ block of the post-nap session in the original Figure. This display error was corrected in Figure 2 of the revised manuscript.

(3) Figure 2b.Showing the main effect of conditions like that is misleading. The reader could interpret it as the difference between non-reactivated and reactivated conditions being significant only for the post-nap offline gains.

We agree with the reviewer that it might indeed be confusing. We altered the legend in Figure 2b and hope that the readability is now increased (see p. 4 in the revised manuscript).

(4) Showing single subject data and the individual change between conditions.– Figure 4 c and d. This is just a suggestion but I think showing individual data points (and the single subject change across the three conditions) would improve understanding the highly significant effects. For example, in Figure 4d the density of SWs for the associated and unassociated condition almost looks the same. However, as this statistical comparison is a within comparison (and relies on the change within a participant) showing the single subject change in addition to the averages per condition increases the readability of the statistical finding.– Figure 2b. Seeing data points and density plots would be helpful especially given the probably skewed distribution for the post-nap offline gains (for the non-reactivated sequence).– same applies to Figure 5.

Individual data points are now presented in the revised manuscript for Figure 2b, Figure 4c and d as well as for Figure 5 (p. 4, p.7, and p. 8 respectively) to better reflect within-subject effects.

(5) Heading titles.I’m not entirely sure whether this is in the realm of the journal, but the Results section might benefit from headings shortly summarising the paragraph (e.g. on page 3. Instead of 2.1. behavioural data you could say something like 2.1. TMR enhanced behavioural performance)

We thank the reviewer for this suggestion. However, we opted to not alter the heading titles for the following two reasons. First, as the Results section reports a large number of analyses (including behavior, ERP, TF, ERPAC, sleep event detection and brain-behavior correlations), we believe that it might be easier for the reader to follow this quite dense section with titles reflecting the type of analysis rather than the main findings. Second, some of the analyses performed under each sub-heading revealed findings that are difficult to summarize in a short title sentence (e.g., results of the brain-behavior correlation analyses). We would therefore prefer keeping the initial titles in order to favor readability.

(6) Figure 3.Using the same colour scheme as in Figure 2 but for different conditions (violet for non-reactivated in Figure 2 and unassociated in Figure 3) is misleading.

We thank the reviewer for bringing this point to our attention and we apologize for this lack of consistency. All figures presenting unassociated conditions (i.e., Figures 3 and 4) have been altered to match the color code introduced in Figure 1 (i.e., the unassociated condition is represented in yellow).

(7) Table 1.To increase the reading flow of the paper and solely focus on the main findings, the authors might want to put table 1 into the supplement. Table 1 contains relevant information to underpin that participants slept and that cues were mainly delivered in NREM sleep but it is not fundamental for the general research question.

Table 1 is now presented in the supplements (Table S1 in Supplementary file 1). We thank the reviewer for this suggestion. We now briefly provide essential information about sleep and stimulation characteristics in the main manuscript.

p. 4, l. 13: “Briefly, results indicate that all the participants slept during the nap (average total sleep time: 67min; average sleep efficiency: 74.9%) and that cues were accurately presented in NREM sleep (average stimulation accuracy: 88.4%).”

(8) No negative peak for unassociated sounds.It is very interesting that the unassociated sounds did not elicit any negative peak. Even though the authors mention this (for me) surprising result in the discussion, I am missing referencing the literature demonstrating an evoked response even to sounds which were not associated with any memory content before (e.g. Cairney et al. (2018), Weigenand(2017)) or which were associated with later forgotten memories (Schreiner et al. (2015)). How do the authors interpret their findings in light of this literature?Cairney SA, Guttesen A á V, El Marj N, Staresina BP. (2018). Memory Consolidation Is Linked to Spindle– Mediated Information Processing during Sleep. Current Biology, 28(6). 948-954.e4.Weigenand et al. (2017). Timing matters: open-loop stimulation does not improve overnight consolidation of word pairs in humans. European Journal of Neuroscience. 45, 629-630.Schreiner, T., Lehmann, M. and Rasch, B. (2015). Auditory feedback blocks memory benefits of cueing during sleep. Nature Communications 6, 8729.

Inspection of the ERP at the individual channel level (cf Supplemental Figure S2) revealed that unassociated auditory cues indeed elicited negative peak on some channels (Fz and C3 to a lesser extent). These results are in line with channel ERP data presented in the supplement of Cairney et al. 2018 (Figure S1), whereby the amplitude of the negative peak fluctuated across channels (see low amplitude of negative peaks for object cues on the left electrodes in particular). However, there are indeed discrepancies between our ERP data and those presented in Weigenand et al., and Schreiner et al. These discrepancies might be explained by various factors such as (i) the sleep stage that was stimulated (e.g., NREM3 in Weigenand et al.); a sleep stage in which N1 amplitudes are usually greater than during NREM2 (Atienza et al., 2001), (ii) the electrode(s) from which ERPs were extracted (e.g., E117 in Schreiner et al.; an electrode that was not included in our montage), (iii) the number of sound repetitions (e.g., 17 repetitions per condition in Schreiner et al., around 195 repetitions per condition in our study). These methodological differences prevent us from making direct comparisons between studies. Nevertheless, these references were added to the revised discussion where the discrepancies highlighted above are now discussed (p. 12, l. 37).

(9) Averaging across all channels.Why do the authors average the data across channels?The ERPs clearly differ across channels (what is explicitly stated in the supplement page 4/5: "The effect reported in the main manuscript across channels is more pronounced on the frontal and the central electrodes" and what is visible in Figure S3)? To argue the averaging in the Results section, they cite Cairney et al. (2018) but I think this is not a convincing justification as Cairney et al. (2018) focused on multivariate analyses where all channels are included in the analyses. However, this is not the case in this study.Further, the TFRs were averaged across channels as well as the data for the phase amplitude coupling. It has been shown that the phase of SOs in which spindles are coupled to differ across the scalp (e.g. Klinzing et al. (2016)).I'm wondering to what extent the channel average conceals for example a SW-spindle coupling during rest which is not visible in Figure 6a but can be expected. The results might be more robust if the analyses were focused on the central region (which can be argued as the task is a motor learning task)?Klinzing, J.G., Mölle, M., Weber, F., Supp, G., Hipp, J.F., Engel, A.K. and Born, J. (2016). Spindle activity phase-locked to sleep slow oscillations. Neuroimage, 134 (2016), pp. 607-616.

We apologize for the lack of justification concerning the averaging procedures. We used two different averaging pipelines. The first one concerns the event-related potential (ERP) and the time-frequency power modulation. We first computed the ERP (or the convolution) at the channel level and then averaged across channels. Data were averaged across channels as the low-density nature of our montage (6 channels) prevented us from performing meaningful topographical analyses. Averaging across channels instead improved signal-to-noise ratio. The second averaging pipeline concerns the event-related phase-amplitude coupling (ERPAC) and computes the across-channel average before the ERPAC computation. We invite the reviewer to consult our response to comment #6 of reviewer #1 for further information and justification about this averaging method. Briefly, this choice was made in order to increase the signal to noise ratio (SNR) as the quality of the phase estimation is particularly sensitive to noise (Gross et al., 2013). Justification for these averaging choices is now provided in the main text (p. 20, l. 25 and p. 21, l. 47).

In addition to these justifications and in order to address the reviewer’s comment on channel level data (as well as similar comments raised by reviewer #1 who was also interested in seeing PAC results at the channel level), we now also report channel level data – as well as corresponding cluster-based permutation tests – for each analysis reported in the main text (i.e., ERPs are shown in Supplemental Figure S2 and S4, correlation between targeted memory reactivation index and power modulation is depicted in Supplemental Figure S7, PAC difference at the negative peak of the SW is in Supplemental Figure S6 and PAC/TMR index correlation in Figure S8). Briefly, channel level data revealed that, as mentioned by the reviewer, central – and to a lesser extent frontal – electrodes mainly contributed to the pattern of results highlighted on averaged maps reported in the main text across the different analyses performed.

Concerning the point on coupling at rest, we kindly refer the reviewer to our response to comment #7 of reviewer #1 and to comment #20 below. Briefly, additional analyses suggest that there is significant SW-sigma coupling at rest despite channel averaging. With respect to the reviewer’s point on focusing on channels located above motor areas, we now present channel level data in the supplements. As hypothesized by the reviewer, channel level data indeed revealed that central electrodes mainly contributed to the pattern of results highlighted with averaged maps and reported in the main text.

Note that we removed the citation of Cairney et al. to justify averaging across channels as channel averaging was indeed only done for a subset of the analyses in this paper

(10) 2.2.1. Event-related analysis. Page 5: "Cluster based permutations tests did not highlight any clusters between the two auditory cues." Here, a CBP was conducted across time to compare unassociated and associated conditions, right? Maybe the authors want to report this result when they talk about the comparison between unassociated and associated conditions (more at the beginning of this paragraph) but this is just a suggestion.

We believe that there might be a misunderstanding here, probably due to a lack of clarity on our side. The sentence quoted by the reviewer refers to the results of the time frequency analyses (oscillatory activity evoked by the cues), not the ERP analyses (potentials evoked by the cues). While we observed a condition effect on ERP negative peak amplitude, there was indeed no effect of condition on oscillatory activity evoked by the cues. We clarified this distinction in the revised manuscript where we now start the event-related section by introducing the different types of analyses (p. 4, l. 17 and p. 5, l. 8) and we further distinguished these different approaches at the start of each corresponding paragraph. We hope this will alleviate the confusion with respect to the event-related analyses.

(11) 2.2.2. Sleep event detection. Page 5 : "[…] see methods for details on the detection algorithms and Table S2 in Supplementary file 2". I believe the info is found in table S4.

We thank the reviewer for pointing out this error. We thoroughly checked the revised manuscript and corrected any discrepancies.

(12) Detection algorithms.Could the authors specify how exactly they detected SWs and spindles? I assume they band passed filtered the signal in the specific frequency range (0.5 – 2 Hz for SWs and 12-16Hz for spindles)? For spindles root mean square was used or was power extracted with the Hilbert transformation? The signal had to exceed a specific threshold for 0.5-3sec to be defined as a spindle? If so, what was the threshold?

We apologize that the sleep event detection methods was not detailed enough in the original manuscript. The information provided below has been added to the revised version of the Materials and methods (p. 21, l. 3 and l. 11).

Concerning spindle detection, the algorithm implemented in YASA (Vallat and Walker, 2021) is largely inspired from the A7 algorithm described in Lacourse et al. (2019). Specifically, the relative power in the spindle frequency band (12-16Hz) with respect to the total power in the broad-band frequency (1–30 Hz) is estimated based on Short-Time Fourier Transforms with 2 s windows and a 200 ms overlap. Next, the algorithm uses a 300 ms window with a step size of 100 ms to compute the moving root mean squared (RMS) of the filtered EEG data in the sigma band. A moving correlation between the broadband signal (1–30 Hz) and the EEG signal filtered in the spindle band is then computed. Sleep spindles are detected when the three following thresholds are reached simultaneously: (1) the relative power in the sigma band (with respect to total power) is above 0.2, (2) the moving RMS crosses the RMS_mean_ + 1.5 RMS_SD_ threshold and (3) the moving correlation described is above 0.65. Additionally, detected spindles shorter than 0.5 s or longer than 2 s were discarded. Spindles occurring on different channels within 500 ms of each other were assumed to reflect the same spindle and were therefore merged together.

Concerning slow-waves (SWs) detection, the algorithm used in YASA is a custom adaptation of the algorithms used in Massimini et al. (2004) and Carrier et al., (2011). Specifically, data are filtered between 0.3 to 2 Hz with a FIR filter with a 0.2 Hz transition resulting in a -6 dB points at 0.2 and 2.1 Hz. Then all the negative peaks with an amplitude between -40 to -200 μV and the positive peaks with an amplitude comprised between 10 to 150 μV are detected in the filtered signal. After sorting identified negative peaks with subsequent positive peaks, a set of logical thresholds are applied to identify the true slow-waves: (1) duration of the negative deflection of the SW between 0.3 to 1.5 sec; (2) duration of the positive deflection of the SW between 0.1 to 1 sec; (3) absolute amplitude of the negative trough of the SW between 40 μV to 300 μV, (4) absolute positive peak amplitude of the SW between 10 μV to 200 μV and (5) peak-to-peak amplitude of the SW between 75 μV to 500 μV.

(13) Page 6, first paragraph: "[…] more importantly, that the associated sounds resulted in an increase in SW amplitude, density and slope as compared to the unassociated sounds." Did the authors compare the slope?

We thank the reviewer for bringing this to our attention. SW slope analyses are not reported in this manuscript. We deleted this information from the revised paper.

(14) Differences in preferred phase between cue-locked and SW-lockedMaybe the authors can plot Table 2 as a rose plot? That way the results would be easier to read. I assume the preferred phase in Table 2 is indicated in radians?Why do the authors see completely different phases of the coupling for the cued locked vs. the spontaneous SWs? Wouldn't you assume to have a similar coupling for evoked responses and spontaneous SWs with spindles?

We would like to sincerely thank the reviewer for raising this point as there was indeed an error in the computation of the SW-locked preferred phases (incorrect time windows were used). We apologize for this. The correct preferred phase values are now presented using rose plots as suggested by the reviewer (Supplemental Figure S5). Yet, to increase readability, these results are now reported in the supplements. Note that results remain unchanged (i.e., no difference between conditions) but that phase values are now indeed more consistent between cue-locked and SW-locked analyses.

(15) Specify how PAC is calculated.In the method section it says: "The strength of the coupling between the phase of the SO signal and the amplitude of the 7-30Hz signal was then computed at each timepoint of the analysis window (every 2ms)." Please state more clearly how the PAC has been calculated and what the PAC factor is (Figure 6a). Did the authors consider using the mean vector length as a measure for the coupling strength?

We acknowledge that this procedure was not detailed enough in the initial manuscript and have added all relevant information and references on p. 21 (l. 47) of the revised manuscript. Briefly, coupling analyses were performed using the Event-Related Phase-Amplitude Coupling (ERPAC) method proposed by Voytek et al. (2013) and implemented in the TensorPac to support multi-dimensional arrays (Combrisson et al., 2020). Unlike blocked PAC that measures PAC across time cycle (e.g. with mean vector length algorithm), ERPAC is calculated across trials separately at each time point (Lachaux et al., 1999) in order to preserve the time dimension. Specifically, the instantaneous phases and envelopes of amplitudes from our two components (slow and sigma oscillations, respectively) were first calculated by Hilbert transform. For each trial time point, we computed the circular-linear correlation of phase and amplitude values across trials. This analysis therefore tested whether trial-by-trial differences in *slow oscillation* phase explained a significant amount of the inter-trial variability in *sigma* amplitude in the analyzed time window. Therefore, the PAC factor represents the correlation coefficient between a circular (slow oscillation phase) and a linear random (sigma amplitude) variable at each time point and across trials and ranges between 0 and 1.

(16) No PAC between SWs and spindles in rest condition.I'm surprised that there seems to be no coupling between the SO phase and the sigma frequencies in the rest condition.

We thank the reviewer for pointing this out and performed additional analyses in order to test for coupling during rest. We invite the reviewer to read our response to comment #7 of reviewer #1 for details. In sum, results highlighted a significant preferred slow oscillation phase for the amplitude peak of sigma in the 3 sec. around the negative peak of the SW occurring during rest (Rayleigh z = 5.7, p-value = 0.003) intervals. These results suggest that there is an inherent SW-sigma coupling, as sigma amplitude consistently peaks at a particular slow oscillation phase. These results are now reported as rose plots in Supplemental Figure S5 and changes have been made in the methods (p. 22, l. 22) and results (p. 8, l. 10, l. 16, and l. 29) sections accordingly.

(17) Figure 6.The color scheme is misleading as in both figures (6a and 6b) the same color scheme is used (ranging from blue to yellow) but the numeric range is different (Figure 6a starts at 0 and is just positively scaled and Figure 6b ranges from -0.025 to 0.05). The authors might want to adapt the color range in Figure 6a.The c-axis (Figure 6b, PAC factor difference) is not symmetric.

We thank the reviewer for these suggestions. We have now changed this figure accordingly.

(18) Figure 6a. unassociated condition.It is noticeable that the PAC is especially high in many frequencies and at different timepoints for the unassociated condition. Of course, a statistical comparison is needed to test that (can be done by a CBP against 0). Do the authors have an explanation for that?

We thank the reviewer for this suggestion. However, the precise analysis outlined by the reviewer would not be overly meaningful with the ERPAC method employed in this study. Notably, as ERPAC values range from 0 to 1, an ERPAC significantly different from zero is essentially a certainty for all points of the time-frequency window. We therefore opted to follow the procedure used in Mikutta et al. (2019) in order test for the presence of coupling during unassociated intervals. We invite the reviewer to read our response to comment #7 of reviewer #1 for details. Briefly, results indeed indicate a significant SW-sigma coupling during unassociated intervals. Note that similar results were observed for associated and rest intervals. Importantly, ERPAC analyses presented in the main text indicate that PAC was significantly stronger around the negative peak of the slow oscillation during unassociated as compared to associated stimulation and rest intervals (see Figure 6b-c in the main text). We speculate that sigma oscillations nested in the trough of the SW during unassociated intervals might prevent the processing of unassociated/irrelevant sounds during post-learning sleep which might in turn be reflected by a decrease in the amplitude of the slow electrophysiological responses (i.e., smaller ERP and SWs) during non-associated sound intervals.

(19) 2.3. Correlation analyses.Page 10.: "[…] Correlation analyses between the TMR index (i.e., the difference in offline gains)" Post-nap or post-night offline gains?

As the behavioral analyses did not reveal any interaction between the condition (reactivated vs. reactivated) and time (post-nap vs. post-night), the TMR index was computed based on changes averaged across the post-nap and post-night intervals. This is now clarified in the revised manuscript (p. 19, l. 29 and p. 9, l. 12).

(20) 2.3. Correlation analyses.Page 10: "[…] that higher TMR index was related to higher sigma oscillation power for the unassociated compared to the associated sound condition" Is it possible that the TMR benefit (higher TMR index) is driven by lower power for the associated cues (potentially due to a stronger evoked response and hence a stronger evoked down-state during which the spindle power in general is lower)?

We thank the reviewer for this interesting suggestion. To test this hypothesis, we ran the cluster-based permutation correlation between the power evoked by the associated cue and the TMR index. This analysis did not highlight any significant cluster (all p-values > 0.1). These results then suggest that the power following the associated cues cannot explain the observed correlation.

(21) FDR correction in results.The authors mentioned in the method section that p-values were corrected using FDR. It would be good to define in the result section whether a p-value was corrected or not.

We now provide in the revised manuscript both un-corrected and corrected p-values whenever correction was performed.

(22) Figure 7. Could you please plot the correlation values between TMR index and TFR power differences as these are the values for the CBP if I understand correctly. Additionally, I'm interested in seeing the TFRs for the associated and unassociated cue locked responses (before taking the difference).

We thank the reviewer for this suggestion and we now provide rho values for each time-frequency representation of the correlation analyses (Figure 7, p. 10 and 8, p. 11 in revised manuscript).

We present in Author response image 2 the TFRs of the power modulation locked to the associated and the unassociated cues. Results show that power modulation was similar between conditions (as described in the main text). Interestingly, power modulation computed across conditions revealed an increase of sigma (and higher frequencies) power from 0.5 to 1 sec post-cue regardless the condition as well as a low frequency increase centered at 0.5 sec. post cue. These results are consistent with the trend of spindle amplitude being higher in the stimulated intervals as compared to rest intervals.

**Author response image 2. sa2fig2:** Time-Frequency Representation (TFR) of group average of the power modulation evoked by the auditory cues averaged across all EEG channels from 5 to 30 Hz (y-axis) from 0 to 2. 5 sec (x-axis) relative to cue onset for the two conditions (left: associated; middle: unassociated) and collapsed across conditions (right). b. Topography of the TFR of the group average collapsed across conditions. Power modulation was significantly higher than zero in response to auditory cues regardless the condition in the highlighted cluster. Red frames indicate the pre-registered sigma frequency band of interest..

(23) Figure 8. Similar to Figure 7, please plot the correlation values as well.

This change has been done.

Reviewer #3 (Recommendations for the authors):– Abstract: The sentence beginning with "Importantly, sounds that were not associated…" is a bit complicated. Perhaps this can be rephrased?

This sentence was broken down in order to increase readability.

“Importantly, sounds that were not associated to learning strengthened SW-sigma coupling at the SW trough. Moreover, the increase in sigma power nested in the trough of the potential evoked by the unassociated sounds was related to the TMR benefit.”

– Behavioral results (page 3): Lower reaction times at the SRTT obviously reflect better performance. Hence, to me, it was puzzling at first to see that the interpretation of offline gains is flipped. It might help to add one short sentence to clarify this.

We added a sentence that will hopefully clarify this point.

p. 3, l. 29: “Post-nap and post-night offline changes in performance were then computed for both conditions as the relative change in speed between the three plateau blocks of the pre-nap test and the first four blocks of the post-nap and post-night sessions, respectively. As such, improvement in performance from training to retest (i.e. faster performance at retest compared to training) was reflected by positive offline changes in performance. A repeated measures analysis of variance …”

– TFR and PAC results (page 5, line 19 and page 8, line 12). Why were the TFR and PAC analysis performed on different lower limits (5 vs. 7 Hz).

The lower limit of the evoked power modulation analyses (5 Hz) was set to obtain the best ratio between temporal and frequential resolution with the duration of our epochs. Concerning the PAC analysis, as we are studying the coupling between the phase of the slow oscillation and the power of the signal in higher frequencies, we wanted to use a more conservative approach with respect to the lower limit in order to avoid the δ frequency range (up to 4 Hz). Considering the leakage of the frequencies at the vicinity of the range of interest when filtering, we set the limit to 7 Hz.

– Page 6, line 20: To some spindle "features" and "characteristics" could mean the same. I would suggest to be more specific and phrase it as "spindle occurrence" or similar.

In this sentence, features and characteristics was intended to be synonymous, as we wanted to highlight that auditory stimulation influenced spindle characteristics but the sound condition did not. However, as it might be misleading, we have now rephrased this sentence as follows:

p. 6, l. 14: “In summary, these results indicate that while auditory stimulation altered spindle features (frequency and amplitude to a lesser extent) as compared to rest, the two sound conditions did not differently influence these characteristics.”

– Figure 3A and 4A/B: Is it possible to move the x-axis of Figure 3A to the bottom and add more labels to the ticks? Otherwise, it is difficult to interpret the timing. The same could be applied to Figure 4A. Moreover, the y-labels for Figure 4B could move a bit to the left.

We thank the reviewer for these suggestions that have been implemented in Figures 3 (p. 5) and 4 (p. 7) of the revised manuscript.

– Color bars: If the color range for the TFR and PAC maps include negative values, I would like to suggest using a symmetric color bar. It would help the reader visualize where Zero lies.

The color bars have been changed accordingly (Figures 6-8 of the revised manuscript).

– Figure 7A: The overlay of significant correlation cluster onto the power difference between associated – unassociated is very confusing. Why didn't the authors directly plot the correlation values? Combining this with a reporting of the TFR plots (see above) would give a much better overview of the important patterns.

We thank the reviewer for this very helpful suggestion. The figures reporting the results of the correlation analyses now depict the rho values instead of the power difference (Figure 7 p. 10) and PAC difference (Figure 8 p. 11 in the revised manuscript).

– Phase-amplitude coupling (page 21): I do not understand how the PAC is computed. It is clear that the 0.5 to 2 Hz signal serves as the phase-reference and is related to power from 7 to 30 Hz in steps of 0.5 Hz. In my view, for each trial the phase of the peak in the related powerband is determined. Circular statistics can then determine if there is a non-uniform coupling and, if so, to which phase it corresponds. However, here the authors state that every 2 ms the strength of coupling is calculated. This means that at every time bin, frequency bin and trial, yields a pair of slow-wave phase and power value. How is the PAC derived from this data? More importantly, SO frequency may differ between subjects and trials, thus I cannot quite grasp how this can be generalized across subjects with a timewise x-axis. I hope the authors understand my confusion and I would appreciate it if they could elaborate on this analysis. Please note also, that a PAC analysis strongly depends on the present power, thus the comparison of the cue-locked conditions might be a confounded difference in slow wave power.

We acknowledge that the methodology describing the PAC analyses was not detailed enough in the initial manuscript and have added all relevant information and references on p. 21 (l. 47) of the revised manuscript. Briefly, coupling analyses were performed using the Event-Related Phase-Amplitude Coupling (ERPAC) method proposed by Voytek et al. (2013) and implemented in the TensorPac to support multi-dimensional arrays (Combrisson et al., 2020). Unlike traditional blocked PAC that measures PAC across time cycle (e.g. with mean vector length algorithm), ERPAC is calculated across trials separately at each time point (Lachaux et al., 1999) in order to preserve the time dimension. Specifically, the instantaneous phases and envelopes of amplitudes from our two components (slow and sigma oscillations, respectively) were first calculated by Hilbert transform. For each trial time point, we computed the circular-linear correlation of phase and amplitude values across trials. This analysis therefore tested whether trial-by-trial differences in *slow oscillation* phase explained a significant amount of the inter-trial variability in *sigma* amplitude in the analyzed time window. Concerning the note of the reviewer on the influence of power on PAC output, we do agree that PAC analyses are sensitive to various factors such as noise and global power. Therefore, the recommendations when performing PAC analysis is to use experimental contrasts in order to subtract out any global power effects (Schoffelen and Gross, 2009). With respect to the potential influence of condition-specific modulation of power, as we did not observe any effect of condition on power (see time-frequency analyses reported in the main text), we believe that it is unlikely that power differences would have confounded the PAC results.

– I found it very interesting that a TMR benefit was found immediately after the nap, which was even improved further after an overnight. This contradicts recent evidence on episodic memory only showing a benefit after the additional overnight (Cairney et al., 2018). Furthermore, while this study uses an unfamiliar sound as a control condition, nap studies are ideal to implement a wake control group. It might be worthwhile to discuss or present as food for thought how the reported results related to episodic memory or what different studies design might bring as insights.

We agree with the reviewer that this point is a valuable addition to the discussion. This is now discussed in the revised manuscript as follow:

p. 12, l. 6: “In the present study, we examined the impact of auditory TMR on motor memory consolidation as well as the neurophysiological processes supporting reactivation during sleep. Our results demonstrate a TMR-induced behavioral advantage such that offline changes in performance were larger on the reactivated as compared to the non-reactivated sequence. These behavioral results are in line with earlier motor learning studies showing improvement in performance after auditory (13, 16, 12) or olfactory (17) TMR during sleep. As opposed to earlier TMR research though, the current results suggest that TMR-induced consolidation is not a protracted process that needs additional time and/or sleep to develop (24), as a behavioral advantage could already be observed immediately after the TMR episode. Also, in contrast to earlier work showing that TMR effects can be transient (12), the current data indicate that the effect of TMR on motor performance was sustained overnight. It remains unclear whether these discrepancies are related to the nature of the task (e.g., declarative vs. motor), the sensory stimulus used for reactivation (words vs. sound) or the duration of the reactivation / sleeping episode (nap vs. night). Nevertheless, our findings suggest that the TMR episode during a nap immediately following learning set the reactivated memory trace on a distinct yet parallel trajectory as compared to the non-reactivated memory trace.”

With respect to the point of the reviewer on a wake control condition, we agree that nap designs are ideally suited to offer such controls. However, we opted to not include such control group in our experiment but to instead include all within-subject control conditions that allowed us to address the two main goals of the present research which were (1) highlighting a behavioral benefit for sequences reactivated during sleep as compared to sequences not reactivated during sleep and (2) investigating the neurophysiological correlates of such TMR-induced benefit. We acknowledge that comparing the neurophysiological processes underlying reactivation during sleep vs. wakefulness is of high interest but this was beyond the scope of the present research.

– In line with my last comment, it is interesting that the authors interpret their findings as opposing facilitating and protective mechanisms mediated by slow waves and sleep spindles, but what are the practical implications? On the one hand, stronger responses upon control cues are beneficial. On the other hand, the slow wave-spindle coupling plays an important role as well, however, in this case for real cues. Does this mean that TMR studies missed out on incorporating a control cue or is it enough to only cue with unfamiliar sounds to protect memories? Is something reactivated during control sounds? Given that control sounds don't really evoke a slow oscillatory response (whereas SW-locked analysis are performed across the while cueing interval) implies that control cue-related mechanisms might emerge after or between cueing.A potential approach to understand all this is multivariate analysis see Cairney et al. 2018; Schreiner et al. 2020 or new preprints from the Lewis lab=. Thus, it might be worthwhile again to discuss the distinct functions of control and real cues with regard to memory reactivation.

We thank the reviewer for this interesting discussion. It appears that the addition of a control sound in our research indeed allowed to highlight these (unexpected) protective processes. Our results indeed suggest that when a control, unknown cue is presented to the sleeping brain, it might trigger protective mechanisms to prevent these “irrelevant” sensory stimuli to be processed and therefore disturb the ongoing consolidation of previously encoded and reactivated memories. Specifically, we speculated that SW-sigma coupling during exposure to unassociated sounds might prevent sound processing which would in turn be reflected by a decrease in the amplitude of the slow electrophysiological responses (i.e., smaller ERP and SWs) during non-associated sound intervals. In order to further examine this possibility and better substantiate this hypothesis, we performed additional exploratory analyses testing for potential relationships between the PAC observed on unassociated conditions and slow electrophysiological responses (i.e., ERP and SWs). To do so, we extracted the PAC value during unassociated stimulation intervals in the time-frequency window where PAC was significantly greater for unassociated as compared to associated and rest conditions (i.e. from -0.5 to 0.5 sec and from 14 to 18 Hz, see Figure 6 in the main text). While the PAC during unassociated intervals did not correlate with the amplitude of the unassociated ERPs, it correlated negatively with the properties of the SWs detected during unassociated intervals. Specifically, the higher the PAC, the lower SW density (t = -2.9, df = 20, p-value = 0.004) and the lower the peak-to-peak SW amplitude (S = 2460, p-value = 0.037) during unassociated intervals (see Supplemental Figure S9 in the revised manuscript). These results provide further support for the protective mechanism discussed above. These correlations are now reported in the supplemental information and mentioned in the revised discussion to further discuss the distinct functions of control and memory cues with regard to memory reactivation.

p. 13, l. 46: “We argue that sigma oscillations might play the role of a gatekeeper for the consolidation process and protect the motor memory trace against potential interfering effects induced by the unassociated sounds which might in turn potentiate the effect of TMR at the behavioral level. In order to further examine this possibility, we performed additional exploratory analyses testing for potential relationships between the SW-sigma coupling observed during unassociated stimulation intervals and slow electrophysiological responses (see Supplemental Figure S9). Results showed a negative correlation between SW-sigma coupling and SW features such that higher coupling was related to lower SW amplitude and density during unassociated stimulation intervals. These results provide further support for the protective effect of sigma oscillations (nested in the trough of slow oscillations) against potential interfering effects induced by the unassociated sounds. These assumptions are also in line with a growing body of literature pointing towards a sensory gating role of spindle activity / sigma oscillations (40, 41) that might be critical to facilitate the memory consolidation process during sleep (42, 39).”

Last, we performed additional exploratory analyses in order to test the interesting hypothesis proposed by the reviewer that control cue-related mechanisms might emerge after or between cueing. To do so, we performed a cluster-based permutation test comparing associated and unassociated cue-locked evoked potentials and oscillations on a broader time window (from -1 to 4 sec relative to cue onset) to cover the entire period in-between cues (inter-stimulus interval of 5 sec.). These analyses did not highlight any significant differences between conditions (ERP: all cluster p-values > 0.2; oscillatory analysis: all cluster p-values > 0.39).

[Editors' note: further revisions were suggested prior to acceptance, as described below.]

The manuscript has been substantially improved and there are solely two methodological remaining issues that need to be addressed, as raised by Reviewer 2 and outlined below regarding the analysis of cross-frequency coupling in the data.

We would like to thank the editor for their time and for their comment on the revised manuscript. We provide below a detailed response to address the remaining issues raised by reviewer 2. We hope that the editor and the reviewers find our responses satisfactory.

Reviewer #2 (Recommendations for the authors):Thanks a lot for a very thorough revision.The majority of my points have been addressed. However, there are still two of my comments (and the authors' responses) I have questions about:Comment #4Thanks for clarifying the phase-amplitude coupling analysis. However, it deviates from the pre-registration, doesn't it? In the pre-registration, the authors describe the coupling analysis as a phase-phase coupling analysis:"SO-spindle coupling: Finally, we aim to determine preferred phase of SO-spindle coupling for both evoked and spontaneous oscillations. We will extract the instantaneous phase of both the SO-filtered signal and of the envelope power in the spindle frequency band. Then we will calculate the circular distance between the phase time series. The preferred phase result from the mean of the circular angle values and will be computed across all trials of each condition separately. "I did not find any justification of that deviation in supplementary file 3. Why did the authors change their analysis approach? Do the results differ?

We thank the reviewer for bringing this issue to our attention which appears to be due to a lack of clarity in the pre-registration. Preferred phase analyses are indeed phase-amplitude and not phase-phase coupling analyses (Canolty et al. (2006); Dupré la Tour et al. (2017); Penny et al. (2008)). We acknowledge that the description of these analyses was confusing in the pre-registration. We indeed extracted the instantaneous phase of the SO-filtered signal and the envelope power in the spindle frequency band. We then computed the preferred phase as the SO phase where sigma amplitude is maximum and represented the mean of the circular angle across trials for each condition in the density plots presented in Figure S5. We hope that this additional information has clarified the analyses conducted.

Comment # 20Thanks a lot to the authors for all their effort to address my concern. However, I still have some concerns when comparing the main with the control analysis:The authors used the procedure by Mikutta et al. (2019) to show that there is SW-sigma oscillation coupling in all three conditions (associated, unassociated, rest). The control analysis is completely valid and compelling to demonstrate SW-sigma oscillation coupling during the rest condition. However, when comparing the main analysis with the control analysis there seems to be some differences:First, there is no difference in the preferred phase angle of the SW-filtered signal when the sigma oscillations peak. Wouldn't you expect a mean phase difference between the associated and unassociated condition given that the coupling is stronger around the SW trough in the unassociated condition (similarly to Figure 6 in the main text)?Second, I would like to see the presence of SW-sigma oscillation coupling with their actual dependent variable (PAC factor) as this is the variable the authors based their findings on. They stated that a statistical comparison to 0 is not a suitable approach as the ERPAC ranges between 0 and 1. Consequently, significant differences between the data and 0 are very likely.An alternative way to create control data (where you don't expect any coupling) is to use events without any SW/cue. For example, you can run the SW detection on your data. For each detected SW you can choose a control event which is a SW free event temporally close to the detected SW (e.g., within 30s pre or post). Based on these SW free events the same analysis as in Figure 6 can be run and statistical comparisons can be made. For a comparable approach see (Ngo et al., 2020).ReferencesNgo, H. V. V., Fell, J., and Staresina, B. (2020). Sleep spindles mediate hippocampal-neocortical coupling during long-duration ripples. ELife, 9, 1-18. https://doi.org/10.7554/eLife.57011

We thank the reviewer for their positive comments about the revised manuscript.

With respect to the comparison between the preferred phase (referred to as control analyses by the reviewer) and the ERPAC results, we did not expect a between-condition difference in ERPAC around the trough of the SO to result in different preferred phases between conditions. Preferred phase measures are independent of ERPAC magnitude. Preferred phase analyses show that sigma power peaked consistently across trials at a similar phase of the SO (descending phase) in both conditions. However, these analyses do not inform on whether – at each time point of the analyzed epoch – inter-trial SO phase variability is correlated to inter-trial sigma amplitude variability. In other words, sigma amplitude might be maximum at a specific phase of the slow oscillation in both conditions (as shown with the preferred phase analyses) but the relationship between the amplitude and the phase of the signal at this particular time point might be different between conditions (as measured with ERPAC). Altogether, the results show that while the SO phase at which sigma peaks was similar between conditions, the across-trial relationship between SO phase and sigma amplitude was stronger in the unassociated as compared to the associated condition in this particular time/SO phase window.

Regarding the additional ERPAC control analyses mentioned by the reviewer, we predict that ERPAC will be significantly different from zero in any within condition analyses for the reason mentioned in the first revision (ERPAC values ranging from 0 to 1). As ERPAC is not suited to test against zero, we therefore opted to use preferred phase (PP) analyses to test for coupling within conditions. Additionally, we expect the SW-free approach suggested by the reviewer will yield significant coupling (even with PP approaches) in analyses windows that are *not* centered on the negative peak of the SO peak as there is general strong coupling between the phase of the 0.5-2 Hz signal (irrespective of whether a SO is formally detected or not) and sigma amplitude during sleep. We performed the suggested SW-free analyses with both ERPAC and PP approaches to support the above-statement. Namely, for each detected SW in the rest condition, we extracted a random time sample from a time window lasting 1 minute around the negative peak of the SW. The random point was not selected if it was part of another SW and we also pseudo-randomly selected the time point in order to obtain a uniform distribution in terms of slow oscillation phase. We first computed the preferred phase of the slow wave at which the sigma oscillations peak and we tested whether the preferred phases were uniformly distributed using Rayleigh test for non-uniformity of circular data (Berens, 2009). In line with our expectations and with the SW-locked analyses, SW-free-locked analyses revealed that the SO phase at which the sigma amplitude was the highest was not distributed uniformly during rest (Rayleigh z = 8.2, p-value = 1.4e-4). SW-free-locked and SW-locked preferred phases did not differ (F(1,43) = 1.4, p-value = 0.24). These results suggest that the coupling between the phase of the SO and the amplitude of the sigma power is not limited to the epochs where SO are detected. Last and as expected from ERPAC testing against zero, both the SW-locked and the SW-free-locked ERPAC analyses show that ERPAC was significantly different from zero at rest.

Another way to create control events is to fill the original trials with random noise (see Combrisson et al., (2020)). Specifically, we created, for each individual, the same number of SW-locked trials as in the original analysis but filled with random noise uniformly distributed between the minimum and the maximum in the signal of each particular participant. Similar as above, we tested the distribution of the resulting preferred phase and ERPAC against zero. As expected, the random noise preferred phase results revealed that the SO phase at which the sigma amplitude was the highest was distributed uniformly (Rayleigh z = 1.6, p-value = 0.21) which indicates the absence of coupling of random noise data. As expected from ERPAC testing against zero, random noise ERPAC was still significantly different from zero.

We hope this additional information better justifies the use of the preferred phase as a control analysis to test for coupling (or its absence) within each condition and at rest as compared to ERPAC testing against zero.